# Demystifying Batch Normalization in ReLU Networks: Equivalent Convex Optimization Models and Implicit Regularization

**Tolga Ergen**,* **Arda Sahiner**,* **Batu Ozturkler, John Pauly, Morteza Mardani & Mert Pilanci**

Department of Electrical Engineering
Stanford University
Stanford, CA 94305, USA
`{ergen,sahiner,ozt,pauly,morteza,pilanci}@stanford.edu`

## Abstract

Batch Normalization (BN) is a commonly used technique to accelerate and stabilize training of deep neural networks. Despite its empirical success, a full theoretical understanding of BN is yet to be developed. In this work, we analyze BN through the lens of convex optimization. We introduce an analytic framework based on convex duality to obtain exact convex representations of weight-decay regularized ReLU networks with BN, which can be trained in polynomial-time. Our analyses also show that optimal layer weights can be obtained as simple closed-form formulas in the high-dimensional and/or overparameterized regimes. Furthermore, we find that Gradient Descent provides an algorithmic bias effect on the standard non-convex BN network, and we design an approach to explicitly encode this implicit regularization into the convex objective. Experiments with CIFAR image classification highlight the effectiveness of this explicit regularization for mimicking and substantially improving the performance of standard BN networks.

## 1 Introduction

Deep neural networks have achieved dramatic progress in the past decade. This dramatic progress largely hinged on improvements in terms of optimization techniques. One of the most prominent recent optimization techniques is Batch Normalization (BN) (Ioffe & Szegedy, 2015). BN is an operation that is introduced in between layers to normalize the output and it has been shown to be extremely effective in stabilizing and accelerating training of deep neural networks. Hence, it became standard in state-of-the-art architectures, e.g., ResNets (He et al., 2016). Despite its empirical success, it is still theoretically elusive why BN is extremely effective for training deep neural networks. Therefore, we investigate the mechanisms behind the success of BN through convex duality.

### 1.1 Related Work

**Batch Normalization:** One line of research has focused on alternatives to BN, such as Layer Normalization (Ba et al., 2016), Instance Normalization (Ulyanov et al., 2016), Weight Normalization (Salimans & Kingma, 2016), and Group Normalization (Wu & He, 2018). Although these techniques achieved performance competitive with BN, they do not provide any theoretical insight about its empirical success.

Another line of research studied the effects of BN on neural network training and identified several benefits. For example, Im et al. (2016) showed that training deep networks with BN reduces dependence on the parameter initialization. Wei et al. (2019) analyzed BN via mean-field theory to quantify its impact on the geometry of the optimization landscape. They reported that BN flattens the optimization landscape so that it enables the use of larger learning rates. In addition, Bjorck et al. (2018); Santurkar et al. (2018); Arora et al. (2018) showed that networks trained with BN achieve

---

*Equal Contribution

faster convergence and generalize better. Furthermore, Daneshmand et al. (2020) proved that BN avoids rank collapse so that gradient-based algorithms, e.g., Stochastic Gradient Descent (SGD), are able to effectively train deep networks. Even though these studies are important to understand the benefits of BN, they fail to provide a theoretically complete characterization for training deep networks with BN.

**Convex Neural Networks:** Recently, a series of papers (Pilanci & Ergen, 2020; Ergen & Pilanci, 2020; 2021a;b; Sahiner et al., 2021a;b) studied ReLU networks through the lens of convex optimization theory. Particularly, Pilanci & Ergen (2020) introduced exact convex representations for two-layer ReLU networks, which can be trained in polynomial-time via standard convex solvers. However, this work is restricted to two-layer fully connected networks with scalar outputs. Later on, Ergen & Pilanci (2021a) first extended this approach to two-layer scalar output Convolutional Neural Networks (CNNs) with average and max pooling and provided further improvements on the training complexity. These results were extended to two-layer fully convolutional networks and two-layer networks with vector outputs (Sahiner et al., 2021a). However, these convex approaches are restricted to two-layer ReLU networks without BN, thus, do not reflect the exact training framework in practice, i.e., regularized deep ReLU networks with BN.

## 1.2 OUR CONTRIBUTIONS

- We introduce an exact convex framework to explicitly characterize optimal solutions to ReLU network training problems with weight-decay regularization and BN. Thus, we obtain closed-form solutions for the optimal layer weights in the high-dimensional and overparameterized regime.
- We prove that regularized ReLU network training problems with BN can be equivalently stated as a finite-dimensional convex problem. As a corollary, we also show that the equivalent convex problems involve whitened data matrices unlike the original non-convex training problem. Hence, using convex optimization, we reveal an implicit whitening effect introduced by BN.
- We demonstrate that GD applied to BN networks provides an implicit regularization effect by learning high singular value directions of the training data more aggressively, whereas this regularization is absent for GD applied to the equivalent whitened data formulation. We propose techniques to explicitly regularize BN networks to capture this implicit regularization effect.
- Unlike previous studies, our derivations extend to deep ReLU networks with BN, CNNs, BN after ReLU[1], vector output networks, and arbitrary convex loss functions.

## 1.3 PRELIMINARIES

**Notation:** We denote matrices and vectors as uppercase and lowercase bold letters, respectively, where a subscript indicates a certain element or column. We use $\mathbf{0}$ (or $\mathbf{1}$) to denote a vector or matrix of zeros (or ones), where the sizes are appropriately chosen depending on the context. We also use $\mathbf{I}_n$ to denote the identity matrix of size $n$. To represent Euclidean, Frobenius, and nuclear norms, we use $\|\cdot\|_2, \|\cdot\|_F$, and $\|\cdot\|_*$, respectively. Lastly, we denote the element-wise 0-1 valued indicator function and ReLU activation as $\mathbb{1}[x \geq 0]$ and $(x)_+ = \max\{x, 0\}$, respectively.

We[2] consider an $L$-layer ReLU network with layer weights $\mathbf{W}^{(l)} \in \mathbb{R}^{m_{l-1} \times m_l}$, where $m_0 = d$ and $m_L = C$ are the input and output dimensions, respectively. Given a training data $\mathbf{X} \in \mathbb{R}^{n \times d}$ and a label matrix $\mathbf{Y} \in \mathbb{R}^{n \times C}$, we particularly focus on the following regularized training problem

$$\min_{\theta \in \Theta} \mathcal{L}(f_{\theta,L}(\mathbf{X}), \mathbf{Y}) + \beta \mathcal{R}(\theta), \tag{1}$$

where we compactly represent the parameters as $\theta := \{\mathbf{W}^{(l)}, \gamma_l, \boldsymbol{\alpha}_l\}_{l=1}^L$ and the corresponding parameter space as $\Theta := \{\{\mathbf{W}^{(l)}, \boldsymbol{\gamma}^{(l)}, \boldsymbol{\alpha}^{(l)}\}_{l=1}^L : \mathbf{W}^{(l)} \in \mathbb{R}^{m_{l-1} \times m_l}, \boldsymbol{\gamma}_l \in \mathbb{R}^{m_l}, \boldsymbol{\alpha}_l \in \mathbb{R}^{m_l}, \forall l \in [L]\}$. We note that $\boldsymbol{\gamma}_l$ and $\boldsymbol{\alpha}_l$ are the parameters of the BN operator, for which we discuss the details below. In addition, $\mathcal{L}(\cdot, \cdot)$ is an arbitrary convex loss function, including squared, hinge, and cross entropy loss, and $\mathcal{R}(\cdot)$ is the regularization function for the layer weights with the tuning parameter $\beta > 0$. We also compactly define the network output as

$$f_{\theta,L}(\mathbf{X}) := \mathbf{A}^{(L-1)}\mathbf{W}^{(L)}, \ \text{ where } \mathbf{A}^{(l)} := \left(\text{BN}_{\gamma,\alpha}\left(\mathbf{A}^{(l-1)}\mathbf{W}^{(l)}\right)\right)_+$$

---

[1]Presented in Appendix G.

[2]All the proofs and some extensions are presented in Appendix.

we denote the $l^{th}$ layer activations as $\mathbf{A}^{(l)} \in \mathbb{R}^{n \times m_l}$, and $\mathbf{A}^{(0)} = \mathbf{X}$. Here, $\text{BN}_{\gamma,\alpha}(\cdot)$ represents the BN operation introduced in Ioffe & Szegedy (2015) and applies matrices column-wise.

**Remark 1.1.** *Note that above we use BN before ReLU activations, which is common practice and consistent with the way introduced in Ioffe & Szegedy (2015). However, BN can be placed after ReLU as well, e.g., Chen et al. (2019), and thus in Section G of Appendix, we will also consider architectures where BN layers are placed after ReLU.*

For a layer with weight matrix $\mathbf{W}^{(l)} \in \mathbb{R}^{m_{l-1} \times m_l}$ and arbitrary batch of activations denoted as $\mathbf{A}_b^{(l-1)} \in \mathbb{R}^{s \times m_{l-1}}$, BN applies to each column $j$ independently as follows

$$\text{BN}_{\gamma,\alpha}\left(\mathbf{A}_b^{(l-1)}\mathbf{w}_j^{(l)}\right) := \frac{(\mathbf{I}_s - \frac{1}{s}\mathbf{1}\mathbf{1}^T)\mathbf{A}_b^{(l-1)}\mathbf{w}_j^{(l)}}{\|(\mathbf{I}_s - \frac{1}{s}\mathbf{1}\mathbf{1}^T)\mathbf{A}_b^{(l-1)}\mathbf{w}_j^{(l)}\|_2}\gamma_j^{(l)} + \frac{\alpha_j^{(l)}\mathbf{1}}{\sqrt{n}}, \tag{2}$$

where $\boldsymbol{\gamma}^{(l)}$ and $\boldsymbol{\alpha}^{(l)}$ are trainable parameters that scale and shift the normalized value. In this work, we focus on the full-batch case, i.e $\mathbf{A}_b^{(l-1)} = \mathbf{A}^{(l-1)} \in \mathbb{R}^{n \times m_l}$. This corresponds to training the network with GD as opposed to mini-batch SGD. We note that our empirical findings with GD indicate identical if not better performance compared to the mini-batch case, which is also consistent with the previous studies Lian & Liu (2019); Summers & Dinneen (2020).

Throughout the paper, we consider a regression framework with squared loss and standard weight-decay regularization. Extensions to general convex loss functions are presented in Appendix B.1. Moreover, below, we first focus on scalar outputs i.e. $C = 1$, and then extend it to vector outputs.

### 1.4 OVERVIEW OF OUR RESULTS

Here, we provide an overview of our main results. To simplify the notation, we consider $L$-layer ReLU networks with scalar outputs, i.e., $m_L = C = 1$ thus the label vector is $\mathbf{y} \in \mathbb{R}^n$, and extend the analysis to vector output networks with the label matrix $\mathbf{Y} \in \mathbb{R}^{n \times C}$ in the next sections.

The regularized training problems for an $L$-layer network with scalar output and BN is given by

$$p_L^* := \min_{\theta \in \Theta} \frac{1}{2}\|f_{\theta,L}(\mathbf{X}) - \mathbf{y}\|_2^2 + \frac{\beta}{2}\sum_{l=1}^{L}\left(\left\|\boldsymbol{\gamma}^{(l)}\right\|_2^2 + \left\|\boldsymbol{\alpha}^{(l)}\right\|_2^2 + \left\|\mathbf{W}^{(l)}\right\|_F^2\right), \tag{3}$$

where we use $\boldsymbol{\gamma}^{(L)} = \boldsymbol{\alpha}^{(L)} = \mathbf{0}$ as dummy variables for notational simplicity.

**Lemma 1.1.** *The problem in (3) is equivalent to the following optimization problem*

$$\min_{\theta \in \Theta_s} \frac{1}{2}\|f_{\theta,L}(\mathbf{X}) - \mathbf{y}\|_2^2 + \beta\left\|\mathbf{w}^{(L)}\right\|_1, \tag{4}$$

*where $\Theta_s := \{\theta \in \Theta : {\gamma_j^{(L-1)}}^2 + {\alpha_j^{(L-1)}}^2 = 1, \forall j \in [m_{L-1}]\}$.*

Using the equivalence in Lemma 1.1, we now take dual of (4) with respect to the output layer weights $\mathbf{w}^{(L)}$ to obtain [3]

$$p_L^* \geq d_L^* := \max_{\mathbf{v}} -\frac{1}{2}\|\mathbf{v} - \mathbf{y}\|_2^2 + \frac{1}{2}\|\mathbf{y}\|_2^2 \text{ s.t. } \max_{\theta \in \Theta_s}\left|\mathbf{v}^\top\left(\text{BN}_{\gamma,\alpha}\left(\mathbf{A}^{(L-2)}\mathbf{w}^{(L-1)}\right)\right)_+\right| \leq \beta. \tag{5}$$

Since the original formulation in (3) is a non-convex optimization problem, any solution $\mathbf{v}$ in the dual domain yields a lower bound for the primal problem, i.e., $p_L^* \geq d_L^*$. In this paper, we first show that strong duality holds in this case, i.e., $p_L^* = d_L^*$, and then derive an exact equivalent convex formulation for the non-convex problem (3). Furthermore, we even obtain closed-form solutions for the layer weights in some cases so that there is no need to train a network in an end-to-end manner.

## 2 TWO-LAYER NETWORKS

In this section, we analyze two-layer networks. Particularly, we first consider a high-dimensional regime, where $n \leq d$, and then extend the analysis to arbitrary data matrices by deriving an equivalent convex program. We also extend the derivations to vector output networks.

---

[3]Details regarding the derivation of the dual problem are presented in Appendix B.3.

We now consider the regularized training problem for a two-layer network with scalar output and BN. Using the equivalence in Lemma 1.1, the problem can be stated as

$$p_2^* = \min_{\theta \in \Theta_s} \frac{1}{2} \left\| f_{\theta,2}(\mathbf{X}) - \mathbf{y} \right\|_2^2 + \beta \left\| \mathbf{w}^{(2)} \right\|_1. \tag{6}$$

We then take dual of (6) with respect to the output weights $\mathbf{w}^{(2)}$ to obtain the following dual problem

$$p_2^* \geq d_2^* = \max_{\mathbf{v}} -\frac{1}{2} \|\mathbf{v} - \mathbf{y}\|_2^2 + \frac{1}{2} \|\mathbf{y}\|_2^2 \text{ s.t. } \max_{\theta \in \Theta_s} \left| \mathbf{v}^T \left( \mathrm{BN}_{\gamma,\alpha} \left( \mathbf{X} \mathbf{w}^{(1)} \right) \right)_+ \right| \leq \beta, \tag{7}$$

where $\Theta_s = \{\theta \in \Theta : \mathbf{w}^{(1)} \in \mathbb{R}^d, \gamma^{(1)^2} + \alpha^{(1)^2} = 1\}$. Notice that here we drop the hidden neuron index $j \in [m_1]$ since the dual constraint scans all possible parameters in the continuous set $\Theta_s$.

## 2.1 High-dimensional Regime ($n \leq d$)

In the sequel, we prove that the aforementioned dual problem is further simplified such that one can find an optimal solution (6) in closed-form as follows.

**Theorem 2.1.** *Suppose $n \leq d$ and $\mathbf{X}$ is full row-rank, then an optimal solution to (6) is*

$$\left( \mathbf{w}_j^{(1)*}, w_j^{(2)*} \right) = \left( \mathbf{X}^\dagger \left( (-1)^j \mathbf{y} \right)_+, (-1)^j \left( \| \left( (-1)^j \mathbf{y} \right)_+ \|_2 - \beta \right)_+ \right)$$

$$\begin{bmatrix} \gamma_j^{(1)*} \\ \alpha_j^{(1)*} \end{bmatrix} = \frac{1}{\| \left( (-1)^j \mathbf{y} \right)_+ \|_2} \begin{bmatrix} \| \left( (-1)^j \mathbf{y} \right)_+ - \frac{1}{n} \mathbf{1} \mathbf{1}^T \left( (-1)^j \mathbf{y} \right)_+ \|_2 \\ \frac{1}{\sqrt{n}} \mathbf{1}^T \left( (-1)^j \mathbf{y} \right)_+ \end{bmatrix}$$

*for $j = 1, 2$. Therefore, strong duality holds, $p_2^* = d_2^*$.*

## 2.2 Exact Convex Formulation

To obtain an equivalent convex formulation to (6), we first introduce a notion of hyperplane arrangements; see also Pilanci & Ergen (2020). We first define a diagonal matrix as $\mathbf{D} := \mathrm{diag}(\mathbb{1}[\mathbf{X} \mathbf{w} \geq 0])$ for an arbitrary $\mathbf{w} \in \mathbb{R}^d$. Therefore, the output of a ReLU activation can be equivalently written as $(\mathbf{X} \mathbf{w})_+ = \mathbf{D} \mathbf{X} \mathbf{w}$ provided that $\mathbf{D} \mathbf{X} \mathbf{w} \geq 0$ and $(\mathbf{I}_n - \mathbf{D}) \mathbf{X} \mathbf{w} \leq 0$ are satisfied. We can define these two constraints more compactly as $(2\mathbf{D} - \mathbf{I}_n) \mathbf{X} \mathbf{w} \geq 0$. We now denote the cardinality of the set of all possible diagonal matrices as $P$, and obtain the following upperbound $P \leq 2r(e(n-1)/r)^r$, where $r := \mathrm{rank}(\mathbf{X}) \leq \min(n, d)$. For the rest of the paper, we enumerate all possible diagonal matrices (or hyperplane arrangements) in an arbitrary order and denote them as $\{\mathbf{D}_i\}_{i=1}^P$ [4]

Based on the notion of hyperplane arrangement, we now introduce an equivalent convex formulation for (6) without an assumption on the data matrix as in the previous section.

**Theorem 2.2.** *For any data matrix $\mathbf{X}$, the non-convex training problem in (6) can be equivalently stated as the following finite-dimensional convex program*

$$\min_{\mathbf{s}_i, \mathbf{s}_i' \in \mathbb{R}^{r+1}} \frac{1}{2} \left\| \sum_{i=1}^P \mathbf{D}_i \mathbf{U}'(\mathbf{s}_i - \mathbf{s}_i') - \mathbf{y} \right\|_2^2 + \beta \sum_{i=1}^P (\|\mathbf{s}_i\|_2 + \|\mathbf{s}_i'\|_2) \text{ s.t. } \begin{matrix} (2\mathbf{D}_i - \mathbf{I}_n)\mathbf{U}'\mathbf{s}_i \geq 0 \\ (2\mathbf{D}_i - \mathbf{I}_n)\mathbf{U}'\mathbf{s}_i' \geq 0 \end{matrix}, \forall i, \tag{8}$$

*where $\mathbf{U} \in \mathbb{R}^{n \times r}$ and $\mathbf{U}' \in \mathbb{R}^{n \times r+1}$ are computed via the compact SVD of the zero-mean data matrix, namely $(\mathbf{I}_n - \frac{1}{n} \mathbf{1} \mathbf{1}^T)\mathbf{X} := \mathbf{U} \mathbf{\Sigma} \mathbf{V}^T$ and $\mathbf{U}' := \begin{bmatrix} \mathbf{U} & \frac{1}{\sqrt{n}} \mathbf{1} \end{bmatrix}$.*

Notice that (8) is a convex program with $2Pr$ variables and $2Pn$ constraints, and thus can be globally optimized via interior-point solvers with $\mathcal{O}(r^6 \left( \frac{n}{r} \right)^{3r})$ complexity.

**Remark 2.1.** *Theorem 2.2 shows that a classical ReLU network with BN can be equivalently described as a convex combination of linear models $\{\mathbf{U}'\mathbf{s}_i\}_{i=1}^P$ and $\{\mathbf{U}'\mathbf{s}_i'\}_{i=1}^P$ multiplied with fixed hyperplane arrangement matrices $\{\mathbf{D}_i\}_{i=1}^P$. Therefore, this result shows that optimal ReLU networks with BN are sparse piecewise linear functions, where sparsity is enforced via the group lasso regularization in (8). The convex program also reveals an implicit mechanism behind BN. Particularly, comparing (8) with the one without BN (see eqn. (8) in Pilanci & Ergen (2020)), we observe that the data matrix for the convex program is whitened, i.e., $\mathbf{U}'^T \mathbf{U}' = \mathbf{I}_{r+1}$.*

---

[4] We provide more details in Appendix B.4.

## 2.3 VECTOR OUTPUT NETWORKS

Here, we analyze networks with vector outputs, i.e., $\mathbf{Y} \in \mathbb{R}^{n \times C}$, trained via the following problem

$$p_{v2}^* := \min_{\theta \in \Theta} \frac{1}{2} \|f_{\theta,2}(\mathbf{X}) - \mathbf{Y}\|_F^2 + \frac{\beta}{2} \sum_{j=1}^{m_1} \left( \left\| \mathbf{w}_j^{(1)} \right\|_2^2 + {\gamma_j^{(1)}}^2 + {\alpha_j^{(1)}}^2 + \left\| \mathbf{w}_j^{(2)} \right\|_2^2 \right). \quad (9)$$

**Lemma 2.1.** *The problem in* (9) *is equivalent to the following optimization problem*

$$p_{2v}^* = \min_{\theta \in \Theta_s} \frac{1}{2} \|f_{\theta,2}(\mathbf{X}) - \mathbf{Y}\|_F^2 + \beta \sum_{j=1}^{m_1} \left\| \mathbf{w}_j^{(2)} \right\|_2, \quad (10)$$

*where* $\Theta_s := \{\theta \in \Theta : {\gamma_j^{(1)}}^2 + {\alpha_j^{(1)}}^2 = 1, \forall j \in [m_1]\}.$

Using the equivalence in Lemma 2.1, we then take dual of (10) with respect to the output layer weights $\mathbf{W}^{(2)}$ to obtain the following dual problem

$$p_{2v}^* \geq d_{2v}^* := \max_{\mathbf{V}} -\frac{1}{2} \|\mathbf{V} - \mathbf{Y}\|_F^2 + \frac{1}{2} \|\mathbf{Y}\|_F^2 \text{ s.t. } \max_{\theta \in \Theta_s} \left\| \mathbf{V}^T \left( \mathrm{BN}_{\gamma,\alpha} \left( \mathbf{X}\mathbf{w}^{(1)} \right) \right)_+ \right\|_2 \leq \beta. \quad (11)$$

**Theorem 2.3.** *The non-convex training problem* (9) *can be cast as the following convex program*

$$p_{v2}^* = \min_{\mathbf{S}_i} \frac{1}{2} \left\| \sum_{i=1}^{P} \mathbf{D}_i \mathbf{U}' \mathbf{S}_i - \mathbf{Y} \right\|_F^2 + \beta \sum_{i=1}^{P} \|\mathbf{S}_i\|_{c_i,*} \quad (12)$$

*for the norm* $\| \cdot \|_{c_i,*}$ *defined as*

$$\|\mathbf{S}\|_{c_i,*} := \min_{t \geq 0} t \text{ s.t. } \mathbf{S} \in t\mathrm{conv}\{\mathbf{Z} = \mathbf{h}\mathbf{g}^T : (2\mathbf{D}_i - \mathbf{I}_n)\mathbf{U}'\mathbf{h} \geq 0, \|\mathbf{Z}\|_* \leq 1\},$$

*where* $\mathrm{conv}\{\cdot\}$ *denotes the convex hull of its argument and* $\mathbf{U}' = \begin{bmatrix} \mathbf{U} & \frac{1}{\sqrt{n}}\mathbf{1} \end{bmatrix}$ *as in Theorem 2.2.*

**Remark 2.2.** *We note that similar to the scalar output case, vector output BN networks training problem involves a whitened data matrix. Furthermore, unlike* (8)*, the nuclear norm regularization in* (12) *encourages a low rank piecewise linear function at the network output.*

The above problem can be solved in $\mathcal{O}((n/d)^d)$ time in the worst case (Sahiner et al., 2021a), which is polynomial for fixed $d$. We note that the problem in (12) is similar to convex semi non-negative matrix factorizations (Ding et al., 2008). Moreover, in certain practically relevant cases, (12) can be significantly simplified so that one can even obtain closed-form solutions as demonstrated below.

**Theorem 2.4.** *Let* $\mathbf{Y}$ *be a one-hot encoded label matrix, and* $\mathbf{X}$ *be a full row-rank data matrix with* $n \leq d$*, then an optimal solution to* (10) *admits*

$$\left( \mathbf{w}_j^{(1)*}, \mathbf{w}_j^{(2)*} \right) = \left( \mathbf{X}^\dagger \mathbf{y}_j, (\|\mathbf{y}_j\|_2 - \beta)_+ \mathbf{e}_j \right), \begin{bmatrix} \gamma_j^{(1)*} \\ \alpha_j^{(1)*} \end{bmatrix} = \frac{1}{\|\mathbf{y}_j\|_2} \begin{bmatrix} \|\mathbf{y}_j - \frac{1}{n}\mathbf{1}\mathbf{1}^T\mathbf{y}_j\|_2 \\ \frac{1}{\sqrt{n}}\mathbf{1}^T\mathbf{y}_j \end{bmatrix}$$

$\forall j \in [C]$*, where* $\mathbf{e}_j$ *is the* $j^{th}$ *ordinary basis vector.*

## 3 CONVOLUTIONAL NEURAL NETWORKS

In this section, we extend our analysis to CNNs. Thus, instead of $\mathbf{X}$, we operate on the patch matrices that are subsets of columns extracted from $\mathbf{X}$ and we denote them as $\mathbf{X}_k \in \mathbb{R}^{n \times h}$, where $h$ is the filter size and $k$ is the patch index. With this notation, a convolution operation between the data matrix $\mathbf{X}$ and an arbitrary filter $\mathbf{z} \in \mathbb{R}^h$ can be represented as $\{\mathbf{X}_k\mathbf{z}\}_{k=1}^K$. Therefore, given patch matrices $\{\mathbf{X}_k\}_{k=1}^K$, regularized training problem with BN can be formulated as follows

$$p_{2c}^* := \min_{\theta \in \Theta_c} \frac{1}{2} \left\| \sum_{j=1}^{m_1} \sum_{k=1}^K \left( \mathrm{BN}_{\gamma,\alpha}^C (\mathbf{X}_k\mathbf{z}_j) \right)_+ w_j - \mathbf{y} \right\|_2 + \frac{\beta}{2} \sum_{j=1}^{m_1} (\|\mathbf{z}_j\|_2^2 + \gamma_j^2 + \alpha_j^2 + w_j^2), \quad (13)$$

where given $\mu_j = \frac{1}{nK}\sum_{k=1}^K \mathbf{1}^T \mathbf{X}_k \mathbf{z}_j$, BN is defined as

$$\mathrm{BN}_{\gamma,\alpha}^C (\mathbf{X}_k \mathbf{z}_j) := \frac{\mathbf{X}_k \mathbf{z}_j - \mu_j \mathbf{1}}{\sqrt{\sum_{k'=1}^K \|\mathbf{X}_{k'} \mathbf{z}_j - \mu_j \mathbf{1}\|_2^2}} \gamma_j^{(1)} + \frac{\alpha_j^{(1)} \mathbf{1}}{\sqrt{nK}}.$$

Using Lemma 1.1, the dual problem with respect to $\mathbf{w}^{(2)}$ is

$$p_{2c}^* \geq d_{2c}^* := \max_{\mathbf{v}} -\frac{1}{2}\|\mathbf{v} - \mathbf{y}\|_2^2 + \frac{1}{2}\|\mathbf{y}\|_2^2 \text{ s.t. } \max_{\theta \in \Theta_s} \left| \mathbf{v}^T \sum_{k=1}^K \left( \mathrm{BN}_{\gamma,\alpha}^C (\mathbf{X}_k \mathbf{z}) \right)_+ \right| \leq \beta. \tag{14}$$

Next, we state the equivalent convex program for CNNs.

**Theorem 3.1.** *The non-convex training problem in* (13) *can be cast as a convex program*

$$\min_{\mathbf{q}_i, \mathbf{q}_i' \in \mathbb{R}^{h+1}} \frac{1}{2} \left\| \sum_{i=1}^{P_c} \sum_{k=1}^K \mathbf{D}_{ik} \mathbf{U}_{Mk}' (\mathbf{q}_i - \mathbf{q}_i') \right\| + \beta \sum_{i=1}^{P_c} (\|\mathbf{q}_i\|_2 + \|\mathbf{q}_i'\|_2) \ s.t. \ \begin{array}{l} (2\mathbf{D}_{ik} - \mathbf{I}_n)\mathbf{U}_{Mk}'\mathbf{q}_i \geq 0 \\ (2\mathbf{D}_{ik} - \mathbf{I}_n)\mathbf{U}_{Mk}'\mathbf{q}_i' \geq 0 \end{array}, \forall i, k \tag{15}$$

*where we first define a new matrix* $\mathbf{M} = [\mathbf{X}_1; \mathbf{X}_2; \ldots \mathbf{X}_K] \in \mathbb{R}^{nK \times h}$ *and then use the following notations: the compact SVD of the zero mean form of this matrix is* $(\mathbf{I}_{nK} - \frac{1}{nK}\mathbf{1}\mathbf{1}^T)\mathbf{M} := \mathbf{U}_M \mathbf{\Sigma}_M \mathbf{V}_M^T$, *and* $\mathbf{U}_M' := \begin{bmatrix} \mathbf{U}_M & \frac{1}{\sqrt{nK}} \end{bmatrix} = [\mathbf{U}_{M1}'; \mathbf{U}_{M2}'; \ldots \ \mathbf{U}_{MK}']$.

Theorem 3.1 proves that (13) can be equivalently stated as a finite-dimensional convex program with $2r_c P_c$ variables (see Remark 3.1 for the definitions) and $2nP_c K$ constraints. Since $P_c$ is polynomial in $n$ and $d$ as detailed in the remark below, we can solve (15) in polynomial-time.

**Remark 3.1.** *The number of arrangements for CNNs rely on the rank of the patch matrices instead of* $\mathbf{X}$. *Particularly, the number of arrangements for* $\mathbf{M}$, *i.e., denoted as* $P_c$, *is upperbounded as* $P_c \leq 2r_c(e(n-1)/r_c)^{r_c}$, *where* $r_c := \mathrm{rank}(\mathbf{M}) \leq h \ll d$. *Therefore, given a fixed filter size* $h$, $P_c$ *is polynomial in* $n$ *and* $d$ *even when* $\mathbf{X}$ *is full rank (see Ergen & Pilanci (2021a) for more details).*

## 4 DEEP NETWORKS

We now analyze $L$-layer neural networks trained via the non-convex optimization problem in (3). Below, we provide an explicit formulation for the last two layers' weights.

**Theorem 4.1.** *Suppose the network is overparameterized such that the range of* $\mathbf{A}^{(L-2)}$ *is* $\mathbb{R}^n$. *Then an optimal solution for the last two layer weights in* (4) *admits*

$$\left( \mathbf{w}_j^{(L-1)*}, w_j^{(L)*} \right) = \left( \mathbf{A}^{(L-2)\dagger} \left( (-1)^j \mathbf{y} \right)_+, (-1)^j \left( \| \left( (-1)^j \mathbf{y} \right)_+ \|_2 - \beta \right)_+ \right)$$

$$\begin{bmatrix} \gamma_j^{(L-1)*} \\ \alpha_j^{(L-1)*} \end{bmatrix} = \frac{1}{\| \left( (-1)^j \mathbf{y} \right)_+ \|_2} \begin{bmatrix} \| \left( (-1)^j \mathbf{y} \right)_+ - \frac{1}{n}\mathbf{1}\mathbf{1}^T \left( (-1)^j \mathbf{y} \right)_+ \|_2 \\ \frac{1}{\sqrt{n}}\mathbf{1}^T \left( (-1)^j \mathbf{y} \right)_+ \end{bmatrix}$$

*for* $j = 1, 2$. *Thus, strong duality holds, i.e.,* $p_L^* = d_L^*$.

### 4.1 VECTOR OUTPUT NETWORKS

Vector output deep networks are commonly used for multi-class classification problems, where one-hot encoding is a prevalent strategy to convert categorical variables into a binary representation that can be processed by neural networks. Although empirical evidence, e.g., Papyan et al. (2020), shows that deep neural networks trained with one-hot encoded labels exhibit some common patterns among classes, it is still theoretically elusive how these patterns emerge. Therefore, in this section, we analyze vector output deep networks trained using one-hot labels via our advocated convex duality.

The following theorem presents a complete characterization for the last two layers' weights.

**Theorem 4.2.** *Let* $\mathbf{Y}$ *be a one-hot encoded label matrix and the network be overparameterized such that the range of* $\mathbf{A}^{(L-2)}$ *is* $\mathbb{R}^n$, *then an optimal solution to* (3) *can be found in closed-form as*

$$\left( \mathbf{w}_j^{(L-1)*}, \mathbf{w}_j^{(L)*} \right) = \left( \mathbf{A}^{(L-2)\dagger} \mathbf{y}_j, (\|\mathbf{y}_j\|_2 - \beta)_+ \mathbf{e}_j \right), \ \begin{bmatrix} \gamma_j^{(L-1)*} \\ \alpha_j^{(L-1)*} \end{bmatrix} = \frac{1}{\|\mathbf{y}_j\|_2} \begin{bmatrix} \|\mathbf{y}_j - \frac{1}{n}\mathbf{1}\mathbf{1}^T \mathbf{y}_j\|_2 \\ \frac{1}{\sqrt{n}}\mathbf{1}^T \mathbf{y}_j \end{bmatrix}$$

$\forall j \in [C]$, *where* $\mathbf{e}_j$ *is the* $j^{th}$ *ordinary basis vector.*

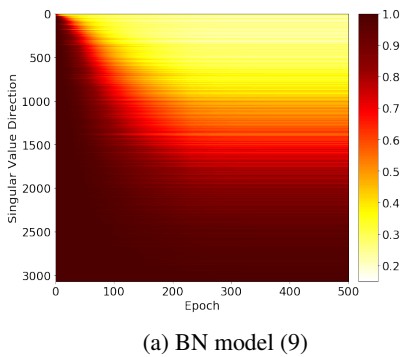

(a) BN model (9)

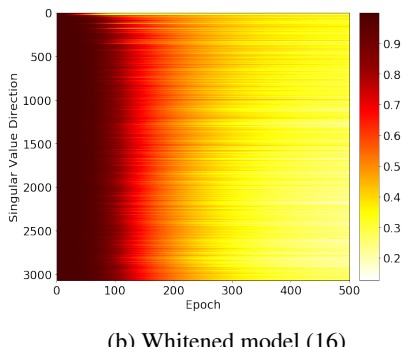

(b) Whitened model (16)

Figure 1: Cosine similarity of network weights relative to initialization for each singular (in descending order) value direction of the data. Here, we use the same setting in Figure 2. We compare training the BN model (9) to the equivalent whitened model (16), noting that the BN model fits high singular value directions of the data first, while the whitened model fits all directions equally, thereby overfitting to non-robust features. This shows implicit regularization effect of SGD.

## 5 IMPLICIT REGULARIZATION OF GD FOR BN

We first note that proposed convex formulations, e.g., (8), (12) and (15), utilize whitened data. Thus, when we train them via a standard optimizers such as GD/SGD, the optimizer might follow an unfavorable optimization path that leads to slower convergence and worse generalization throughout the training. Based on this, we first analyze the impact of utilizing a whitened data matrix for the non-convex problem in (9) trained with GD. Then, we characterize this impact as an implicit regularization due to the choice of optimizer. Finally, we propose an approach to explicitly incorporate this benign implicit regularization into both convex (8) and non-convex (9) training objectives.

### 5.1 EVIDENCE OF IMPLICIT REGULARIZATION

**Lemma 5.1.** *The non-convex problem* (9) *is equivalent to the following problem (i.e., $p_{v2}^* = p_{v2b}^*$)*

$$p_{v2b}^* := \min_{\theta \in \Theta} \frac{1}{2} \left\| \sum_{j=1}^{m_1} \left( \frac{\mathbf{U}\mathbf{q}_j}{\|\mathbf{q}_j\|_2} \gamma_j^{(1)} + \frac{\mathbf{1}}{\sqrt{n}} \alpha_j^{(1)} \right)_+ \mathbf{w}_j^{(2)T} - \mathbf{Y} \right\|_F^2 + \frac{\beta}{2} \sum_{j=1}^{m_1} \left( \gamma_j^{(1)2} + \alpha_j^{(1)2} + \left\| \mathbf{w}_j^{(2)} \right\|_2^2 \right). \tag{16}$$

(16) is equivalent to a problem with new data matrix $\mathbf{U}$ and weight normalization (Salimans & Kingma, 2016). The identity in Lemma 5.1 is achieved by substituting the transformation $\mathbf{q}_j = \mathbf{\Sigma}\mathbf{V}^\top \mathbf{w}_j^{(1)}$.

While (9) and (16) have identical global optima, in practice, GD behaves quite differently when applied to the two problems. Furthermore, these differences extend to the convex formulations in (8), (12), and (15). In particular, while these programs are adept at finding the global optimum, they generalize quite poorly when trained with local search algorithms such as GD. This suggests an implicit regularization effect of GD, the form of which is demonstrated in the following theorem.

**Theorem 5.1.** *Consider* (9) *and* (16) *with corresponding models $f_{v2}(\mathbf{X})$ and $f_{v2b}(\mathbf{U})$. At iteration $k$ of GD, assume that $f_{v2}^k(\mathbf{X}) = f_{v2b}^k(\mathbf{U})$, i.e. both models have identical weights, besides $\mathbf{q}_j^{(k)} = \mathbf{\Sigma}\mathbf{V}^\top {\mathbf{w}_j^{(1)}}^{(k)}$. Then, the subsequent GD updates to model weights are identical, besides those to $\mathbf{w}_j^{(1)}$ and $\mathbf{q}_j$. In $\mathbf{q}_j$-space, the updates $\Delta_{v2,j}^{(k)}$ and $\Delta_{v2b,j}^{(k)}$, respectively, admit*

$$\Delta_{v2,j}^{(k)} = \mathbf{\Sigma}^2 \Delta_{v2b,j}^{(k)}.$$

**Remark 5.1.** *Theorem 5.1 indicates that when solving* (16)*, GD will fit directions corresponding to higher singular values of data much slower relative to other directions compared to* (9)*. By fitting these directions more aggressively, GD applied to* (9) *provides an implicit regularization effect that fits to more robust features. We also note that although our analysis holds only for GD, we empirically observe and validate the emergence of this phenomenon for networks trained with SGD as well.*

To verify this intuition, in Figure 2, we compare the training and test loss curves of solving both (9) and (16) with mini-batch SGD applied to CIFAR-10 (Krizhevsky et al., 2014), demonstrating that SGD applied to (9) generalizes much better than (16). In Figure 1, we plot the cosine similarity of the weights of both networks compared to their initialization in each singular value direction.

Our results confirm that the BN model fits high singular value directions much faster than low ones, and in fact the lower-singular value directions almost do not move at all from their initialization. In contrast, the whitened model fits all singular value directions roughly equally, meaning that it overfits to less robust components of the data. These experiments confirm our theoretical intuition that SGD applied to BN networks has an implicit regularization effect.

## 5.2 Making Implicit Regularization Explicit

Since the convex programs (8), (12), and (15) rely on the whitened data matrix $\mathbf{U}$, they lack the regularization necessary for generalization when optimized with GD. Thus, we aim to impose an explicit regularizer which does not hamper the performance of standard BN networks, but can improve the generalization of the equivalent programs trained using first-order optimizers such as GD. One simple idea to prevent the whitened model from fitting low singular value directions is to obtain a low-rank approximation of the data by filtering its singular values:

$$g(\mathbf{X}) := \mathbf{U}\mathcal{T}(\boldsymbol{\Sigma})\mathbf{V}^\top, \tag{17}$$

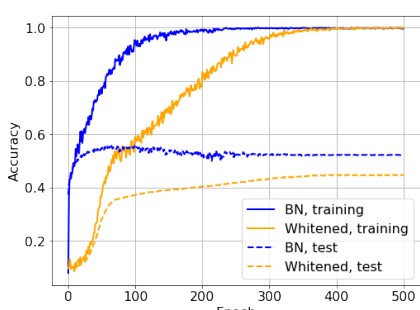

Figure 2: CIFAR-10 classification accuracies for a two-layer ReLU network in (9) and the equivalent whitened model (16) with SGD and $(n, d, m_1, \beta, \mathrm{bs}) = (50000, 3072, 1000, 10^{-4}, 1000)$, where bs denotes batch size. The whitened model overfits much more than the standard BN model, indicating implicit regularization.

with $\mathcal{T}_k(\boldsymbol{\Sigma})_{ii} := \sigma_i \mathbb{1}\{i \geq k\}$, which takes the $k \leq r$ top singular values and removes the rest. We can then solve (1) using $f_{\theta, L}(g(\mathbf{X}))$ as the network output, to form what we denote as the truncated problem. This can also reduce the problem dimension in the whitened-data case since one can simply omit the columns of $\mathbf{U}$. We find this low-rank approximation to be effective, which can even improve the generalization of GD applied to BN networks.

## 6 Numerical Experiments

Here[5], we present experiments to verify our theory. Throughout this section, we denote the baseline model (3) and our non-convex model with truncation as "SGD" and "SGD-Truncated", respectively. Likewise, the proposed convex models are denoted as "Convex" and "Convex-Truncated".

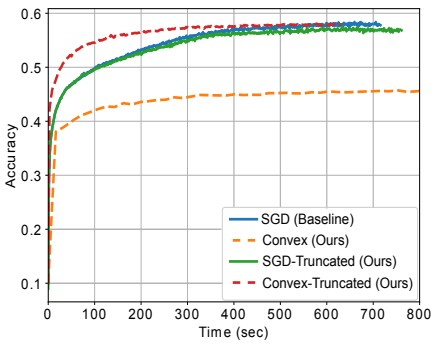
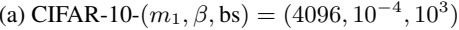

(a) CIFAR-10-$(m_1, \beta, \mathrm{bs}) = (4096, 10^{-4}, 10^3)$

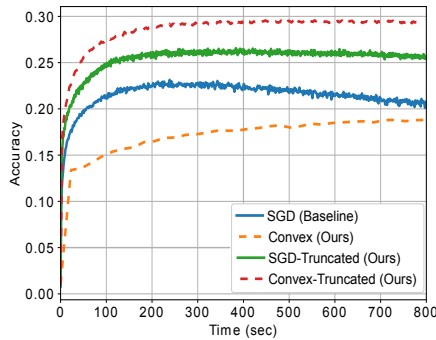

(b) CIFAR-100-$(m_1, \beta, \mathrm{bs}) = (1024, 10^{-4}, 10^3)$

Figure 3: Test accuracy of two-layer BN ReLU networks including the standard non-convex (baseline), its convex equivalent (8), along with their truncated versions with $k = (215, 200)$ respectively. All problems are solved with SGD. The learning rates are chosen based on test performance and we provide learning rate and optimizer comparisons in Appendix A.

---

[5]We present additional numerical experiments and details on the experiments in Appendix A.

**Closed-Form Solutions:** We first verify Theorem 2.4. We consider a three-class classification task with the CIFAR-100 image classification dataset (Krizhevsky et al., 2014) trained with squared loss and one-hot encoded labels, while we set $(n, d, m_1, \beta) = (1500, 3072, 1000, 1)$, where bs denotes the batch size. In Figure 4, we compare the objective values achieved by the closed-form solution and the classical network (6). We observe that our formula is computationally light-weight and can quickly find a global optimum that cannot be achieved with GD even after convergence.

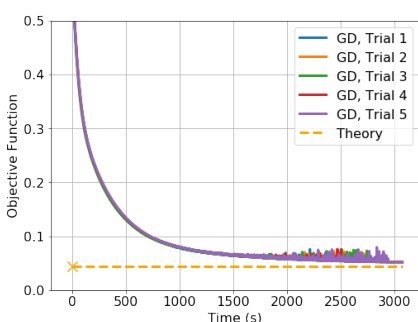

Figure 4: Comparing the objective values of the closed-form solution in Theorem 2.4, ("Theory"), and GD (5 trials) applied to the non-convex problem (9) for three-class CIFAR classification with $(n, d, m_1, \beta) = (1500, 3072, 1000, 1)$. Here, "x" denotes the computation time for the closed-form solution.

**Convex Formulation and Implicit Regularization:** We now demonstrate the effectiveness of our convex formulation to solve (9) on CIFAR-10 and CIFAR-100 (Krizhevsky et al., 2014). Here, instead of enumerating all hyperplane arrangements, we randomly sample via randomly generated Gaussian vectors such that $m_1 = P$. We compare the results of training with and without the explicit regularizer (17) for both cases. We particularly choose $k$ such that singular values explain 95% of the variance in the training data. Then, we consider two-layer networks for image classification on CIFAR-10 with $(n, d, m_1, \beta, \text{bs}, k, C) = (5\text{x}10^4, 3072, 4096, 10^{-4}, 10^3, 215, 10)$ and CIFAR-100 with $(n, d, m_1, \beta, \text{bs}, k, C) = (5\text{x}10^4, 3072, 1024, 10^{-4}, 10^3, 200, 100)$. In Figure 3, we observe that Convex generalizes poorly (as expected with our results in Section 5.1). After the singular value truncation, SGD, SGD-Truncated, and Convex-Truncated generalize equally well for CIFAR-10, and Convex-Truncated in fact generalizes better for CIFAR-100. This shows that the optimizer benefits from the convex landscape of the proposed formulation (8). Furthermore, truncation actually improves the performance of SGD in the case of CIFAR-100, suggesting this is a good regularizer to use.

## 7 CONCLUDING REMARKS

In this paper, we studied training of deep weight-decay regularized ReLU networks with BN, and introduced a convex analytic framework to characterize an optimal solution to the training problem. Using this framework, we proved that the non-convex training problem can be equivalently cast as a convex optimization problem that can be trained in polynomial-time. Moreover, the convex equivalents involve whitened data matrix in contrast to the original non-convex problem using the raw data, which reveals a whitening effect induced by BN. We also extended this approach to analyze different architectures, e.g., CNNs and networks with BN placed after ReLU. In the high dimensional and/or overparameterized regime, our characterization further simplifies the equivalent convex representations such that closed-form solutions for optimal parameters can be obtained. Therefore, in these regimes, there is no need to train a network in an end-to-end manner. In the light of our results, we also unveiled an implicit regularization effect of GD on non-convex ReLU networks with BN, which prioritizes learning high singular-value directions of the training data. We also proposed a technique to incorporate this regularization to the training problem explicitly, which is numerically verified to be effective for image classification problems. Finally, we remark that after this work, similar convex duality arguments were applied to deeper networks Ergen & Pilanci (2021c;d;e); Wang et al. (2021) and Wasserstein Generative Adversarial Networks (WGANs) Sahiner et al. (2022). Extensions of our convex BN analysis to these architectures are left for future work.

### ACKNOWLEDGMENTS

This work was partially supported by the National Science Foundation under grants ECCS- 2037304, DMS-2134248, the Army Research Office.

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

# Appendix

## Table of Contents

## A  ADDITIONAL DETAILS AND NUMERICAL EXPERIMENTS

In this section, we present the details about our experimental setup in the main paper and provide new numerical experiments.

We first note that since we derive exact convex formulations in Theorem 2.2, 2.3, 3.1, we can use solver such as CVX (Grant & Boyd, 2014) and CVXPY (Diamond & Boyd, 2016; Agrawal et al., 2018) with the SDPT3 solver (Tütüncü et al., 2001) to train regularized ReLU networks training problems with BN. Although these solvers ensure quite fast convergence rates to a global minimum, they do not scale to moderately large datasets, e.g., CIFAR-10. Therefore, in our main experiments,

we use an equivalent unconstrained form for the constrained convex optimization problems such that it can be simply trained via SGD. Particularly, let us consider the constrained convex optimization problem in (8), for which the constraints can be incorporated into the objective as follows

$$\min_{\mathbf{s}_i, \mathbf{s}'_i} \frac{1}{2} \left\| \sum_{i=1}^{P} \mathbf{D}_i \mathbf{U}'(\mathbf{s}_i - \mathbf{s}'_i) - \mathbf{y} \right\|_2^2 + \beta \sum_{i=1}^{P} (\|\mathbf{s}_i\|_2 + \|\mathbf{s}'_i\|_2)$$
$$+ \rho \sum_{i=1}^{P} \mathbf{1}^T \left( (-(2\mathbf{D}_i - \mathbf{I}_n)\mathbf{U}'\mathbf{s}_i)_+ + (-(2\mathbf{D}_i - \mathbf{I}_n)\mathbf{U}'\mathbf{s}'_i)_+ \right)$$

where $\rho > 0$ is a hyper-parameter to penalize the violating constraints. Therefore, using the unconstrained form above, we are able to use SGD or other standard first-order optimizers to optimize the constrained convex optimization problem in (8).

Unless otherwise stated, all problems are run on a single Titan Xp GPU with 12 GB of RAM, with the Pytorch deep learning library (Paszke et al., 2019). All SGD models use a momentum parameter of 0.9. All baseline models are trained with standard weight-decay regularization as outlined in (3). All batch-norm models are initialized with scale $\gamma_j^{(l)} = 1$ and bias $\alpha_j^{(l)} = 0$, and all other non-convex model weights are initialized with standard Kaiming uniform initialization (He et al., 2015). For the convex program, we simply initialize all weights to zero.

## A.1 INFERENCE IN BN MODELS

For the non-convex standard BN network, we perform inference in the traditional way. All BN problems with SGD take an exponential moving average of the mini-batch mean and standard deviation with a momentum of 0.1, as is used as a default in deep learning packages, except in the case of full-batch GD, in which case a momentum value of 1 is used. In truncated variants, while the training data is truncated with the explicit regularizer $g(\mathbf{X})$, for test data $\hat{\mathbf{X}}$, we compute the standard forward pass $f_{\theta,L}(\hat{\mathbf{X}})$, i.e. without changing the test data in any fashion, and removing the training-data mean and normalizing by the training-data standard deviation as found with the exponential moving average method described above. For the convex program, we can form the weights to the non-convex standard BN network from the convex program weights, and compute the test-set predictions with the non-convex standard BN network.

## A.2 ADDITIONAL EXPERIMENT: BATCH SIZES FOR SGD

The model of BN we present in this paper is with the model with GD without mini-batching. In this section, we demonstrate that mini-batching does not change the training or generalization performance of BN networks in any significant manner. Our observations also align with previous work examining this subject, such as Lian & Liu (2019); Summers & Dinneen (2020).

In particular, we compare the effect of different batch sizes using a four layer CNN, where the first three layers are convolutional, followed by BN, ReLU, and average pooling, and the final layer is a fully-connected layer. Each CNN layer has 1000 filters with kernel size $3 \times 3$ and padding of 1.

We take the first two classes from CIFAR-100 to perform a binary classification task with $\mathbf{X} \in \mathbb{R}^{1000 \times 3072}$ and $\mathbf{y} \in \mathbb{R}^{1000}$. For $(\text{lr}, \beta) = (10^{-6}, 10^{-4})$, where lr denotes the learning rate, and weight-decay is applied to all parameters. We compare the effect of using a batch size of $50, 100, 500$, and $1000$. We run the batch size 1000 case for 501 epochs, and then train the other batch size cases such that the training time is roughly matched. Due to memory constraints, the full-batch form of SGD does not fit on a standard GPU–thus, all models are trained with a standard CPU with 256 GB of RAM. We compare the train and test accuracy curves in Figure 5, where we see that all batch sizes perform roughly equivalently in both test and training performance. This demonstrates that our full-batch model for BN accurately captures the same dynamics as BN as used in the mini-batch case.

We note that we also perform an ablation study on batch sizes for the nonconvex programs in Section 6 in Appendix A.7.2, and find the same conclusion.

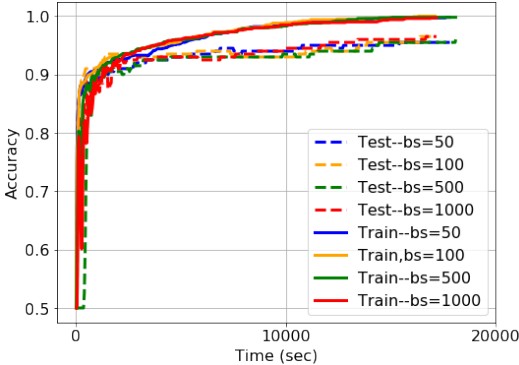

Figure 5: Comparison of different batch sizes (bs) for four-layer CNN with BN layers, $(m_l, \text{lr}, \beta) = (1000, 10^{-6}, 10^{-4})$. We demonstrate that different batch sizes perform roughly equivalently in both training and test convergence.

### A.3  ADDITIONAL EXPERIMENT: SINGULAR VALUE TRUNCATION ABLATION

In order to make clear the effect of the singular value truncation as proposed in Section 5.2 on generalization performance, we include an additional ablation study on a new dataset to verify our intuition. In particular, we evaluated the proposed methods on the CNAE-9 dataset (Ciarelli & Oliveira, 2009), a dataset of 1080 emails with 856 text features over 9 classes. Following Ciarelli & Oliveira (2009), the first 900 samples were used as the training set, while the final 180 samples are used as the test set in this experiment, to obtain $\mathbf{X} \in \mathbb{R}^{900 \times 856}$, $\mathbf{y} \in \mathbb{R}^{900 \times 9}$.

We evaluated the proposed approaches with $(m_1, \beta, \text{bs}) = (10000, 10^{-2}, 900)$, comparing the baseline non-convex SGD approach to the convex truncated formulation for $k \in \{10, 100, 200, 400, 856\}$, where $k$ is the index for which all smaller singular values are truncated from the training dataset. We trained each model for roughly an equal amount of time (15 seconds). For all architectures, a learning rate of $10^{-5}$ was fixed.

We find in Figure 6, that as expected, singular value truncation can effectively match the training and test performance of the SGD BN baseline, but only in a certain range of $k$. When $k$ is too low, such as $k = 10$, valuable features of the data are eliminated and the convex program underfits, whereas when $k$ is too large, such as $k = 856$, the convex program overfits and does not generalize well. Values of $k = 100$ and $k = 200$ generalize the best for this problem, and their corresponding truncations preserve $80\%$ and $91\%$ of the training data respectively. The exact value of $k$ which is optimal for generalization is a hyper-parameter that can be tuned, though it is clear that a large range of values can often work well. This ablation lends additional support to the truncation approach proposed in Section 5.2.

### A.4  ADDITIONAL EXPERIMENT: CONVOLUTIONAL ARCHITECTURES

To demonstrate the effectiveness of the convex formulation in convolutional settings, we provide an additional experiment to show the success of this approach. In particular, as the baseline architecture, we consider a two-layer network which consists of a convolutional layer, followed by a channel-wise BN, followed by a ReLU activation, then subsequently followed by an average pooling operation, flattening, and a fully-connected layer. To use our convex formulation (e.g. (15)) for such an architecture, we can split an input image into spatial blocks (each block corresponds to a region whose outputs will be averaged), and apply a global-average pooling model as in (15) to each spatial block separately and combine the predictions from each spatial block together to obtain one prediction. This suffices as a relaxation of the baseline convolutional architecture.

To this end, we implement the non-convex and convex architectures on CIFAR-10 with the AdamW optimizer for 50 epochs. The convolution is with a $3 \times 3$ kernel with padding of 2 and stride of 3, with average pooling to a spatial dimension of $4 \times 4$. The parameters cho-

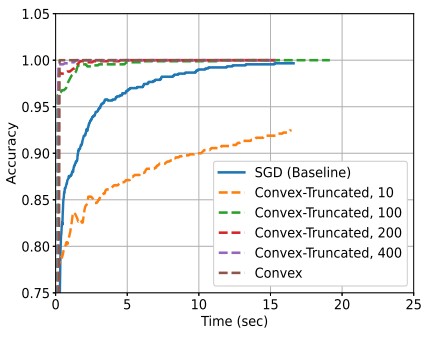

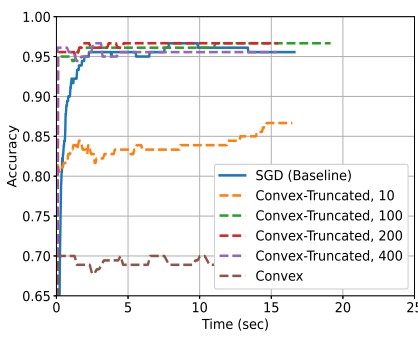

(a) CNAE-9-Training accuracy

(b) CNAE-9-Test accuracy

Figure 6: Comparison of different truncation schemes of convex program compared to SGD baseline, with $(m_1, \beta, \mathrm{bs}, \mathrm{lr}) = (10000, 10^{-2}, 900, 10^{-5})$, and $k \in \{10, 100, 200, 400, 856\}$, where $k = 856$ is simply the convex program without truncation. Large values of $k$ overfit, while low values underfit, and intermediate values match exactly the performance of SGD baseline.

| Method | Train Accuracy | Test Accuracy |
|---|---|---|
| Baseline | 87.33 | 67.06 |
| Baseline Truncated | 78.41 | 66.62 |
| Convex (Ours) | 81.60 | 66.91 |
| Convex Truncated (Ours) | 81.35 | **67.11** |

Table 1: Results of CIFAR-10 classification with two-layer ReLU CNN architectures, with a $3 \times 3$ kernel size, padding of 2, stride of 3, and averaging to $4 \times 4$ spatial dimensions, with $(n, m_1, \beta, \mathrm{bs}, k, C, \mathrm{lr}) = (5 \times 10^4, 1024, 5 \times 10^{-4}, 250, 22, 10, 10^{-2})$. Consistent with our results on fully-connected architectures, the convex architecture matches or improves upon the performance of the non-convex architecture, verifying our theoretical results. We also observe the less pronounced impact of truncation in convolutional experiments.

sen for this model are given by $(n, m_1, \beta, \mathrm{bs}, k, C, \mathrm{lr}) = (5 \times 10^4, 1024, 5 \times 10^{-4}, 250, 22, 10, 10^{-2})$.

We compare the results of the baseline, baseline-truncated, convex, and convex-truncated approaches. In Table 1, we see that the accuracy of this CNN approach is higher than the fully-connected methods we reported in our paper, and as expected, there is a strong correspondence between the performance of the convex and baseline models, again reiterating our strong theoretical results, even in the case of this convex relaxation.

As illustrated in Table 1, truncation enables the convex approach to obtain equivalent or higher test accuracy than its non-convex counterparts, while obtaining similar training accuracy. However, notice that the accuracy improvement provided by truncation is considerably less than the ones for fully-connected layers (e.g. Figure 3a). This phenomenon is essentially due the distribution of the singular values of the data matrix. Particularly, for fully connected layers, we have $\mathbf{U} \in \mathbb{R}^{n \times d}$, and therefore the dimension we truncate has $d = 3072$ features. In this case, the distribution of singular values follows an exponentially decaying pattern such that the ratio of maximal and minimum singular values, also known as the condition number of the data, is quite large ($\sigma_{\max}/\sigma_{\min} = 5892.5$). On the contrary, for CNNs, we operate on the patch matrices so that $\mathbf{U}_{mk} \in \mathbb{R}^{n \times h}$, where $h = 27$ is the size of each convolutional filter. Since the data matrix is much more well conditioned ($\sigma_{\max}/\sigma_{\min} = 271.2$) and has a significantly smaller feature dimension, the impact of truncation is less emphasized in our experiment.

### A.5 EXPERIMENTS FROM SECTION 5.1–EVIDENCE OF IMPLICIT REGULARIZATION

As described in the main paper, for both the standard BN architecture and whitened architecture, we use the parameters $(n, d, m_1, \beta, \mathrm{bs}, \mathrm{lr}) = (50000, 3072, 1000, 10^{-4}, 1000, 10^{-4})$, where bs stands for batch size and lr stands for learning rate. Both models were trained with SGD for 501 epochs,

and the learning rate was decayed by a factor of 2 in the case of a training loss plateau. We now expand on the definition of cosine similarity to clarify Figure 1 (in the main paper).

For weights $(\mathbf{W}, \mathbf{W}')$, we define the distance measure

$$d_{\mathbf{X},i}(\mathbf{W}, \mathbf{W}') := \frac{(\mathbf{v}_i^\top \mathbf{W})(\mathbf{v}_i^\top \mathbf{W}')^\top}{\|\mathbf{v}_i^\top \mathbf{W}\|_2 \|\mathbf{v}_i^\top \mathbf{W}'\|_2} \tag{18}$$

where $\mathbf{v}_i$ are the columns of $\mathbf{V}$ from the SVD of the zero-mean data $(\mathbf{I}_n - \frac{1}{n}\mathbf{1}\mathbf{1}^\top)\mathbf{X}$. This distance metric thus computes the cosine similarity between $\mathbf{W}$ and $\mathbf{W}'$ after being projected to the right singular subspace $\mathbf{v}_i$.

For the same networks trained in Figure 2 (in the main paper), we compute $\{d_{\mathbf{X},i}(\mathbf{W}^{(1)^{(k)}}, \mathbf{W}^{(1)^{(0)}})\}_{i=1}^d$ for the BN model (9), and $\{d_{\mathbf{U},i}(\mathbf{Q}^{(k)}, \mathbf{Q}^{(0)})\}_{i=1}^d$ for the equivalent whitened model (16), for each epoch $k$. We display these in Figure 1 (in the main paper). These correspond to a measure of how far the hidden-layer weights move from their initial values over time in the right singular subspace of the data, with lower values indicating that the weights have moved further from their initialization.

## A.6  VERIFICATION OF CLOSED-FORM SOLUTIONS

See Figure 4 for the results of this experiment. The GD baseline model is trained with a learning rate of $10^{-5}$, and the learning rate was decayed by a factor of 2 in the case of a training loss plateau. The closed-form solution takes only 0.578 seconds to compute, whereas we run the GD baseline model for 40001 epochs, which takes approximately 50 minutes to run per trial. Each trial behaves slightly differently due to the randomness of the initialization of each network.

In terms of the generalization performance, the closed-form expression obtains a test three-class classification accuracy of 47.7%, whereas the baseline GD networks achieve an average of 60.1% final test accuracy, again indicating the implicit regularization effect imposed by GD applied to BN networks.

## A.7  EXACT CONVEX FORMULATION AND IMPLICIT REGULARIZATION

In this section, we describe additional details about our main experimental results of the main paper. In particular, we show the effect of training all models for a longer duration than displayed in the main paper, the different learning rates considered, the different batch sizes considered, along with the effect of using Adam as opposed to SGD, and comparison to a non-BN architecture.

For these experiments, we trained each model (SGD, SGD-Truncated, Convex, Convex-Truncated) for roughly the same amount of time, and considered a range of learning rates for each problem. In particular, for CIFAR-10, we considered the learning rates $\{10^{-5}, 5 \times 10^{-5}, 10^{-4}\}$, and trained the SGD, SGD-Truncated, Convex, and Convex-Truncated models for 501, 501, 51, and 201 epochs respectively, and each model was trained for roughly 700 seconds. In contrast, for CIFAR-100, we considered the learning rates $\{10^{-5}, 5 \times 10^{-5}, 10^{-4}\}$, and trained the SGD, SGD-Truncated, Convex, and Convex-Truncated models for 621,1001,41, and 251 epochs respectively, and each model was trained for roughly 800 seconds.

The range of learning rates was chosen such that all models reached their peak test accuracy within the allotted time, and the learning rate with the best final test accuracy was chosen for each model. In particular for CIFAR-10, the chosen learning rates for the SGD, SGD-Truncated, Convex, and Convex-Truncated models were $10^{-5}$, $10^{-5}$, $5 \times 10^{-5}$, and $10^{-4}$, respectively. In contrast, for CIFAR-100 the chosen learning rates for the SGD, SGD-Truncated, Convex, and Convex-Truncated models were $5 \times 10^{-5}$, $10^{-4}$, $10^{-4}$, and $5 \times 10^{-4}$, respectively.

For our CIFAR-10 experiments in this section, we always use $(m_1, \beta, \text{bs}) = (4096, 10^{-4}, 10^3)$, while for CIFAR-100, we used $(m_1, \beta, \text{bs}) = (1024, 10^{-4}, 10^3)$, where bs stands for batch size, as described in the main paper. The only exception is during our batch size ablation study in Section

A.7.2, in which case the batch size is varied. All models are trained with SGD, except in Section A.7.4.

### A.7.1 TRAINING CURVES AND OVERFITTING

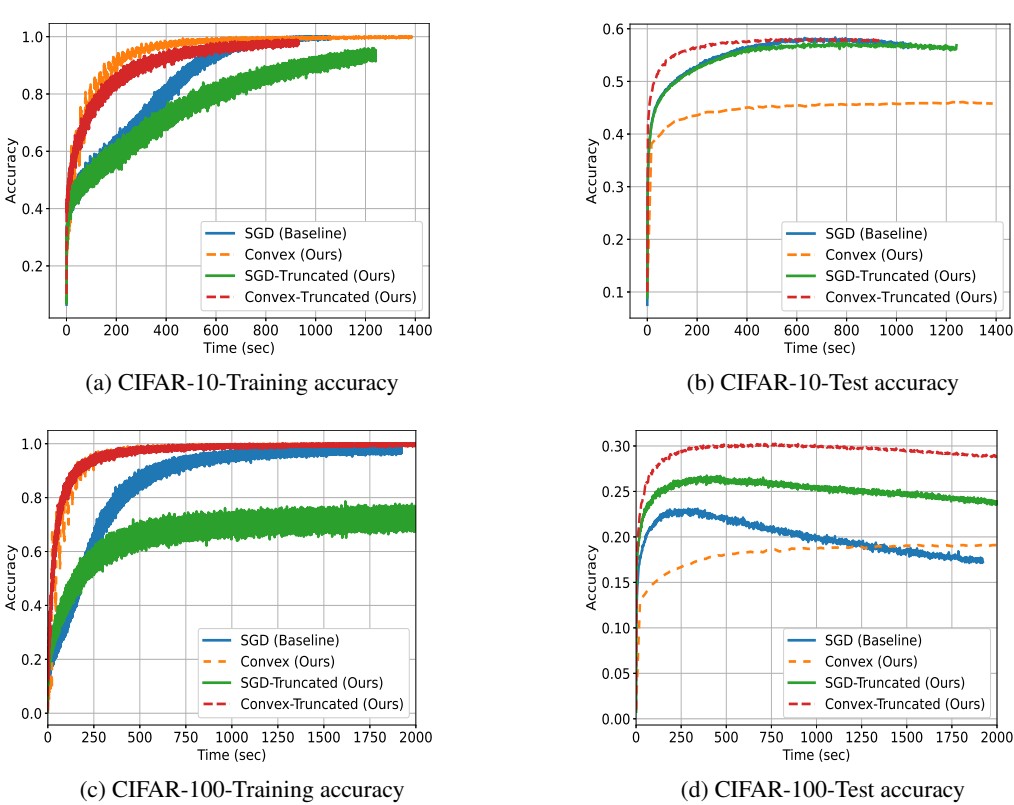

(a) CIFAR-10-Training accuracy

(b) CIFAR-10-Test accuracy

(c) CIFAR-100-Training accuracy

(d) CIFAR-100-Test accuracy

Figure 7: Here, we provide a longer trained version of the CIFAR-10 and CIFAR-100 experiments in Figure 3 (in the main paper). We see that training for longer does not affect the results, as all models begin to overfit, and they overfit at the same rate. For CIFAR-100, we observe that the baseline overfits so much that the standard Convex model eventually overtakes it in generalization performance.

To make more robust our claims in the main paper, we consider training each network for a longer duration, such that each network can near 100% training accuracy. We plot the results of both training and test accuracy for CIFAR-10 and CIFAR-100 in Figure 7. We note that extending training for longer only seems to increase over-fitting, and increasing train accuracy to 100% does not improve model performance. Therefore, the results of our main paper, which display the curves for a shorter duration, represent the peak performance of all algorithms before they begin to overfit.

### A.7.2 IMPACT OF DIFFERENT BATCH SIZES

To further bolster our claim that taking the convex dual form from a full-batch case of SGD is sufficient for capturing the performance of standard BN networks, we compare the performance of Convex and Convex-Truncated with SGD with different batch sizes, namely bs $\in \{100, 500\}$. In Figure 8, we show that different batch sizes do not improve the generalization performance of BN networks, thereby validating that the convex dual derived from the full-batch model is accurate at representing BN networks when trained with mini-batch SGD. Therefore, for all of our other experiments, we use bs $= 1000$.

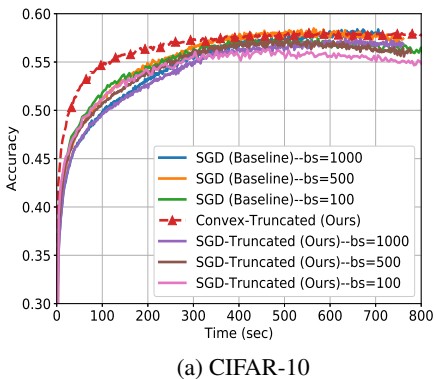
(a) CIFAR-10

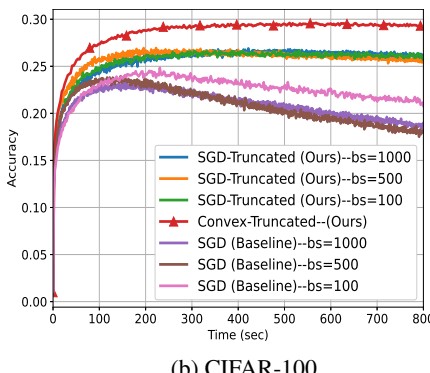
(b) CIFAR-100

Figure 8: Batch size (bs) ablation study, considering bs $\in \{100, 500, 1000\}$ for SGD (Baseline) and SGD-Truncated for the experiment in Figure 3 (in the main paper). We see that smaller batch sizes do not improve the performance of SGD. Therefore, we choose bs $= 1000$ for all the main experiments.

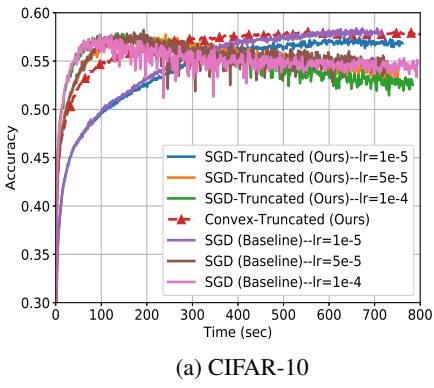
(a) CIFAR-10

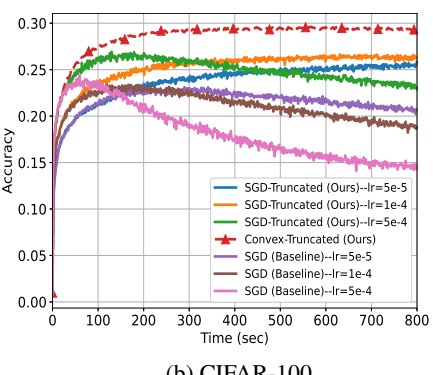
(b) CIFAR-100

Figure 9: Learning rate (lr) ablation study for the experiment in Figure 3 (in the main paper), considering lr $\in \{10^{-5}, 5 \times 10^{-5}, 10^{-4}\}$ for SGD and SGD-Truncated. We see that larger lrs do not improve the performance of SGD, because these models with larger lrs reach the same peak accuracy but overfit quicker. For CIFAR-10, we selected learning rates $10^{-5}$ for both SGD and SGD-Truncated. For CIFAR-100, we chose learning rates $5 \times 10^{-5}$ for SGD, and $10^{-4}$ for SGD-Truncated.

### A.7.3 IMPACT OF DIFFERENT LEARNING RATES

We plot test curves for all considered learning rates for each program for the baseline program for CIFAR-10 and CIFAR-100 in Figure 9. We see that the different learning rates considered do not significantly change the peak performance, with higher learning rates simply overfitting earlier.

### A.7.4 IMPACT OF DIFFERENT OPTIMIZERS

We also consider training both the standard non-convex and our convex formulations with the Adam optimizer (Kingma & Ba, 2014). Not surprisingly, we see a similar implicit regularization effect when applied to BN networks. For the Adam experiments, we consider the same set of learning rates as the SGD experiments, and fix hyper-parameters $(\beta_1, \beta_2, \epsilon) = (0.9, 0.999, 10^{-8})$, as are the standard default values. We run each program for the number of epochs as the SGD experiments, and consider the same learning rates as the SGD experiments. For CIFAR-10, the chosen learning rates for all the models was $10^{-5}$. For CIFAR-100, the chosen learning rates for the Adam, Adam-Truncated, Convex, and Convex-Truncated models were $10^{-5}$, $10^{-4}$, $5 \times 10^{-5}$, and $10^{-5}$, respectively.

We plot the results of the test curves in Figure 10, and plots of longer trained curves in Figure 11, as well as the effect of different learning rates in Figure 12. We find that with Adam, the results mirror that of SGD. For the best learning rate chosen for each method, the truncated convex formulation performs the best, followed by the truncated non-convex and standard non-convex

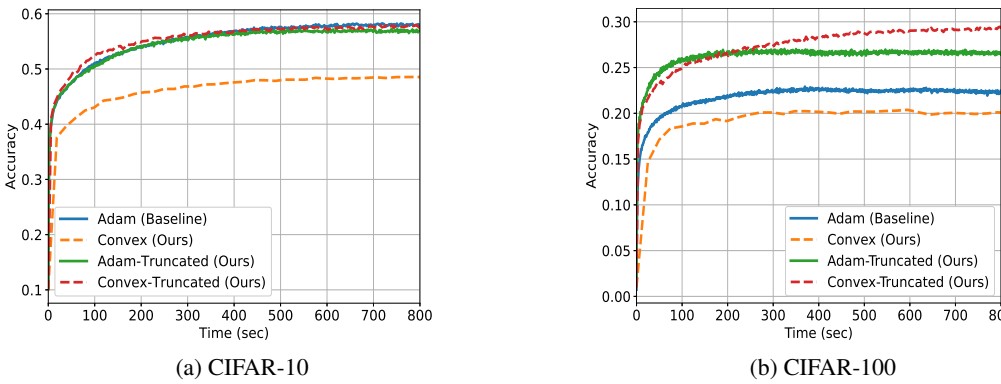

(a) CIFAR-10

(b) CIFAR-100

Figure 10: A counterpart of the experiment in Figure 3 (in the main paper), where we show test accuracies and all problems are solved with Adam. We see that these results mirror those with SGD, where Convex-Truncated performs equally well as or outperforms the baseline.

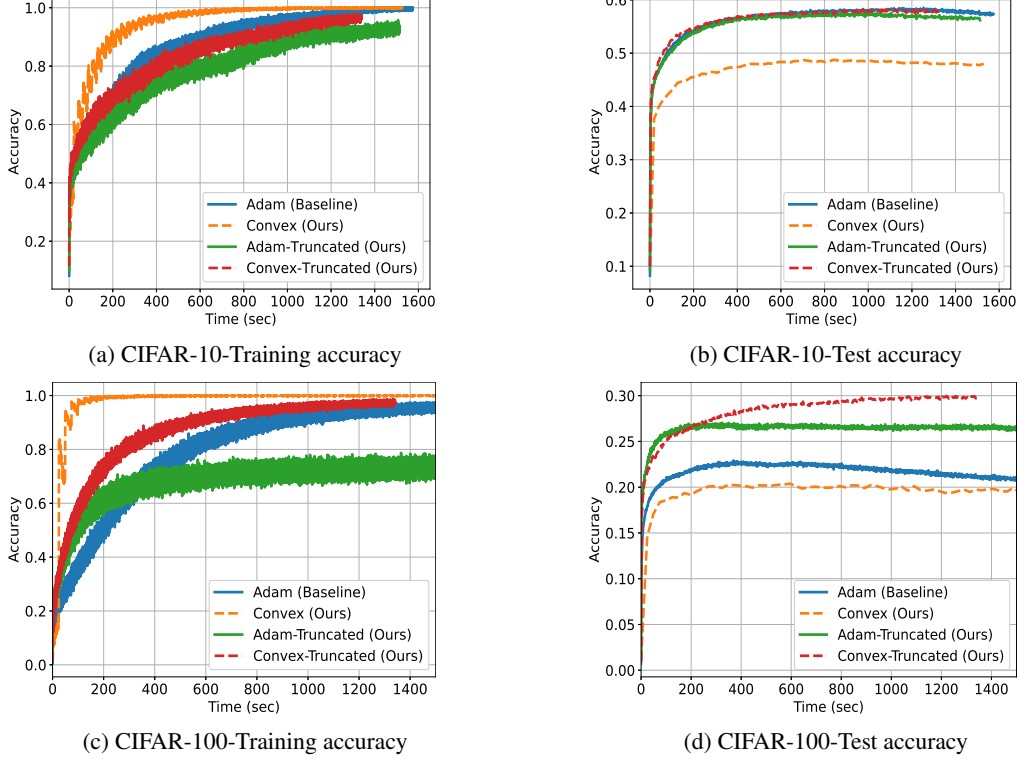

(a) CIFAR-10-Training accuracy

(b) CIFAR-10-Test accuracy

(c) CIFAR-100-Training accuracy

(d) CIFAR-100-Test accuracy

Figure 11: Here, we provide a longer trained version of the CIFAR-10 and CIFAR-100 experiments in Figure 10. We see that training for longer does not change the results, as all models begin to overfit.

formulations, followed by the standard convex formulation. We can conclude that our results are robust to the choice of optimizer.

### A.7.5 COMPARISON OF BN TO NON-BN TWO-LAYER ARCHITECTURES

One might speculate what the impact of adding BN to a two-layer network can have, compared to a simple standard fully-connected architecture. In this section, we compare the standard two-layer network in both its primal and convex dual form, as first presented in Pilanci & Ergen (2020), to the BN networks described in this paper. We let all parameters, i.e. $(n, d, m_1, \beta, \mathrm{bs}, C)$ be identical to the setting of the main paper, and again consider a sweep of learning rates as was done for the BN experiments. For the non-convex primal no BN form, the best performance came from a learning rate

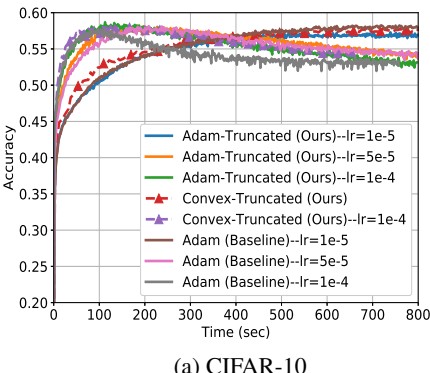 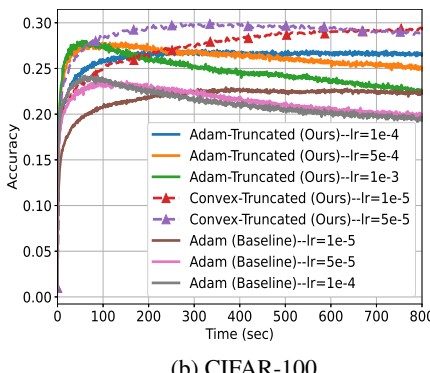

(a) CIFAR-10          (b) CIFAR-100

Figure 12: Learning rate (lr) ablation study for the experiment in Figure 10, where we consider lr $\in$ $\{10^{-5}, 5 \times 10^{-5}, 10^{-4}\}$ for Adam. We see that larger learning rates do not improve the performance of Adam or Convex-Truncated, because these models reach the same peak accuracy but simply overfit earlier. For all models, we select a learning rate of $10^{-5}$ for the CIFAR-10 experiment in Figure 10a. We choose a learning rate of $10^{-5}$ for Adam and Convex-Truncated, and a learning rate of $10^{-4}$ for Adam-Truncated for the CIFAR-100 experiment in Figure 10b.

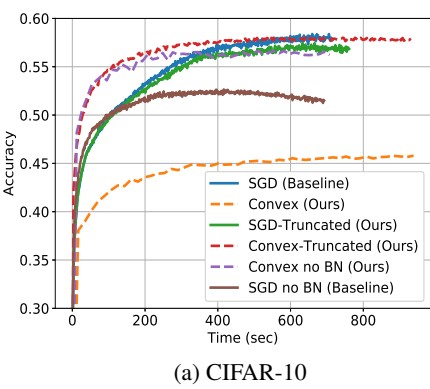 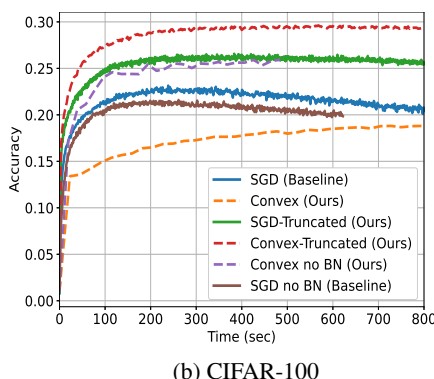

(a) CIFAR-10          (b) CIFAR-100

Figure 13: Comparison of two-layer BN networks presented Figure 3 (in the main paper), to the non-convex two-layer ReLU architecture without BN, and its equivalent convex dual as presented in Pilanci & Ergen (2020). All problems are solved with SGD and the learning rates are chosen based on test performance. We find that both the SGD and Convex-Truncated programs improve in generalization performance when adding BN, though the margin may vary depending on the problem. We also observe that the Convex BN program without truncation does not perform as well as the standard no-BN architecture.

of $5 \times 10^{-7}$ on CIFAR-10 and $10^{-5}$ for CIFAR-100, while for the convex no BN problem, the best performance was with a learning rate of $5 \times 10^{-10}$ on CIFAR-10 and $5 \times 10^{-9}$ for CIFAR-100.

We then compare the best results of each method in Figure 13. We see that for both CIFAR-10 and CIFAR-100, introducing a BN layer can significantly improve the performance of the two-layer network architecture, both when compared to the result found with SGD, and the equivalent convex formulation.

# B  ADDITIONAL THEORETICAL RESULTS

In this section, we present additional theoretical results that are not included in the main paper due to the page limit.

### B.1 Extensions to Arbitrary Convex Loss Functions

In this section, we prove that our convex characterization holds for arbitrary convex loss functions including cross entropy and hinge loss. We first restate two-layer training problem with arbitrary convex loss functions after the resealing in Lemma 1.1

$$\min_{\theta \in \Theta_s} \mathcal{L}(f_{\theta,L}(\mathbf{X}), \mathbf{y}) + \beta \left\| \mathbf{w}^{(L)} \right\|_1 \tag{19}$$

where $\mathcal{L}(\cdot, \mathbf{y})$ is a convex loss function. Then, the dual of (19) is given by

$$\max_{\mathbf{v}} -\mathcal{L}^*(\mathbf{v}) \text{ s.t. } \max_{\theta \in \Theta_s} \left| \mathbf{v}^T \mathrm{BN}_{\gamma,\alpha} \left( \mathbf{A}^{(L-2)} \mathbf{w}^{(1)} \right) \right| \leq \beta,$$

where $\mathcal{L}^*$ is the Fenchel conjugate function defined as Boyd & Vandenberghe (2004)

$$\mathcal{L}^*(\mathbf{v}) = \max_{\mathbf{z}} \mathbf{z}^T \mathbf{v} - \mathcal{L}(\mathbf{z}, \mathbf{y}).$$

The analysis above proves that our convex characterization in the main paper applies to arbitrary loss function. Therefore, optimal parameters for (3) are a subset of the set that includes the extreme points of the dual constraint in (5) independent of loss function.

### B.2 Proof of Lemma 1.1

Here, we first prove that regularizing the intermediate BN parameters and hidden layer weights don't affect the optimization problem therefore the set of optimal solutions for the problems in (20) and (22) are the same. Then, we show that (22) can also be cast as an optimization problem with $\ell_1$ norm regularization penalty on the output layer weights $\mathbf{w}_L$.

#### B.2.1 Effective Regularization in (3)

Here, we prove our claim about the effective regularization. Let us restate the training problem as follows

$$p_{Lr}^* := \min_{\theta \in \Theta} \frac{1}{2} \left\| \sum_{j=1}^{m_{L-1}} \left( \mathrm{BN}_{\gamma,\alpha} \left( \mathbf{A}^{(L-2)} \mathbf{w}_j^{(L-1)} \right) \right)_+ w_j^{(L)} - \mathbf{y} \right\|_2^2 + \frac{\beta}{2} \sum_{l=1}^{L} \left( \left\| \boldsymbol{\gamma}^{(l)} \right\|_2^2 + \left\| \boldsymbol{\alpha}^{(l)} \right\|_2^2 + \left\| \mathbf{W}^{(l)} \right\|_F^2 \right), \tag{20}$$

where we use $\boldsymbol{\gamma}^{(L)} = \boldsymbol{\alpha}^{(L)} = \mathbf{0}$ as dummy variables for notational simplicity. Now, let us rewrite (20) as

$$p_{Lr}^* = \min_{t \geq 0} \min_{\theta \in \Theta} \frac{1}{2} \left\| \sum_{j=1}^{m_{L-1}} \left( \mathrm{BN}_{\gamma,\alpha} \left( \mathbf{A}^{(L-2)} \mathbf{w}_j^{(L-1)} \right) \right)_+ w_j^{(L)} - \mathbf{y} \right\|_2^2 + \frac{\beta}{2} \sum_{j=1}^{m_{L-1}} \left( \gamma_j^{(L-1)^2} + \alpha_j^{(L-1)^2} + w_j^{(L)^2} \right) + \frac{\beta}{2} t$$

$$\text{s.t. } \sum_{l=1}^{L-2} \left( \left\| \boldsymbol{\gamma}^{(l)} \right\|_2^2 + \left\| \boldsymbol{\alpha}^{(l)} \right\|_2^2 + \left\| \mathbf{W}^{(l)} \right\|_F^2 \right) + \left\| \mathbf{W}^{(L-1)} \right\|_F^2 \leq t.$$

After applying the scaling in Lemma 1.1, we then take the dual with respect to $\mathbf{w}^{(L)}$ as in the main paper to obtain the following problem

$$p_{Lr}^* \geq d_{Lr}^* := \max_{t \geq 0} \max_{\mathbf{v}} -\frac{1}{2} \|\mathbf{v} - \mathbf{y}\|_2^2 + \frac{1}{2} \|\mathbf{y}\|_2^2 + \frac{\beta}{2} t \tag{21}$$

$$\text{s.t. } \max_{\theta \in \Theta_{sr}} \left| \mathbf{v}^T \left( \mathrm{BN}_{\gamma,\alpha} \left( \mathbf{A}^{(L-2)} \mathbf{w}^{(L-1)} \right) \right)_+ \right| \leq \beta.$$

where $\Theta_{sr} := \{ \theta \in \Theta : \gamma_j^{(L-1)^2} + \alpha_j^{(L-1)^2} = 1, \forall j \in [m_{L-1}], \sum_{l=1}^{L-2} \left( \left\| \boldsymbol{\gamma}^{(l)} \right\|_2^2 + \left\| \boldsymbol{\alpha}^{(l)} \right\|_2^2 + \left\| \mathbf{W}^{(l)} \right\|_F^2 \right) + \left\| \mathbf{W}^{(L-1)} \right\|_F^2 \leq t \}$. Since

$$\mathrm{BN}_{\gamma,\alpha} \left( \mathbf{A}^{(L-2)} \mathbf{w}_j^{(L-1)} \right) = \underbrace{\frac{(\mathbf{I}_n - \frac{1}{n} \mathbf{1} \mathbf{1}^T) \mathbf{A}^{(L-2)} \mathbf{w}_j^{(L-1)}}{\| (\mathbf{I}_n - \frac{1}{n} \mathbf{1} \mathbf{1}^T) \mathbf{A}^{(L-2)} \mathbf{w}_j^{(L-1)} \|_2}}_{\mathbf{h}(\theta')} \gamma_j^{(L-1)} + \frac{1}{\sqrt{n}} \alpha_j^{(L-1)},$$

where $\theta'$ denotes all the parameters except $\boldsymbol{\gamma}^{(L-1)}, \boldsymbol{\alpha}^{(L-1)}, \mathbf{w}^{(L)}$. Then, independent of the value $t$, $\mathbf{h}(\theta')$ is always a unit norm vector. Therefore, the maximization constraint in (21) is independent of the norms of the parameters in $\theta'$. Consequently, regularizing the weights in $\theta'$ does not affect the dual characterization in (21).

Based on this observation, (20) can be equivalently stated as

$$p_L^* := \min_{\theta \in \Theta} \frac{1}{2} \|f_{\theta,L}(\mathbf{X}) - \mathbf{y}\|_2^2 + \frac{\beta}{2} \sum_{j=1}^{m_{L-1}} \left( \gamma_j^{(L-1)^2} + \alpha_j^{(L-1)^2} + w_j^{(L)^2} \right) \tag{22}$$

### B.2.2 RESCALING FOR $\ell_1$ NORM REGULARIZATION IN (4)

We first note that similar approaches are also presented in Neyshabur et al. (2014); Savarese et al. (2019); Pilanci & Ergen (2020); Ergen & Pilanci (2020); **?**.

For any $\theta \in \Theta$, we can rescale the parameters as $\bar{\gamma}_j^{(L-1)} = \eta_j \gamma_j^{(L-1)}$, $\bar{\alpha}_j^{(L-1)} = \eta_j \alpha_j^{(L-1)}$ and $\bar{w}_j^{(L)} = w_j^{(L)}/\eta_j$, for any $\eta_j > 0$. Then, the network output becomes

$$\begin{aligned}
f_{\bar{\theta},L}(\mathbf{X}) &= \sum_{j=1}^{m_{L-1}} \left( \frac{(\mathbf{I}_n - \frac{1}{n}\mathbf{1}\mathbf{1}^T)\mathbf{A}^{(L-2)}\mathbf{w}_j^{(L-1)}}{\|(\mathbf{I}_n - \frac{1}{n}\mathbf{1}\mathbf{1}^T)\mathbf{A}^{(L-2)}\mathbf{w}_j^{(L-1)}\|_2} \bar{\gamma}_j^{(L-1)} + \mathbf{1}\bar{\alpha}_j^{(L-1)} \right)_+ \bar{w}^{(L)} \\
&= \sum_{j=1}^{m_{L-1}} \left( \frac{(\mathbf{I}_n - \frac{1}{n}\mathbf{1}\mathbf{1}^T)\mathbf{A}^{(L-2)}\mathbf{w}_j^{(L-1)}}{\|(\mathbf{I}_n - \frac{1}{n}\mathbf{1}\mathbf{1}^T)\mathbf{A}^{(L-2)}\mathbf{w}_j^{(L-1)}\|_2} \gamma_j^{(L-1)} + \mathbf{1}\alpha_j^{(L-1)} \right)_+ w^{(L)} \\
&= f_{\theta,L}(\mathbf{X}),
\end{aligned}$$

which proves $f_{\theta,2}(\mathbf{X}) = f_{\bar{\theta},2}(\mathbf{X})$. Moreover, we use the following inequality

$$\frac{1}{2} \sum_{j=1}^{m_{L-1}} \left( \gamma_j^{(L-1)^2} + \alpha_j^{(L-1)^2} + w_j^{(L)^2} \right) \geq \sum_{j=1}^{m_{L-1}} \left| w_j^{(L)} \right| \sqrt{\gamma_j^{(L-1)^2} + \alpha_j^{(L-1)^2}},$$

where the equality is achieved when

$$\eta_j = \left( \frac{|w_j^{(L)}|}{\sqrt{\gamma_j^{(L-1)^2} + \alpha_j^{(L-1)^2}}} \right)^{\frac{1}{2}}$$

is used. Since this scaling does not alter the right-hand side of the inequality, without loss of generality, we can set $\sqrt{\gamma_j^{(L-1)^2} + \alpha_j^{(L-1)^2}} = 1, \forall j \in [m_{L-1}]$. Therefore, the right-hand side becomes $\|\mathbf{w}^{(L)}\|_1$ and (22) is equivalent to

$$\min_{\theta \in \Theta_s} \frac{1}{2} \|f_{\theta,L}(\mathbf{X}) - \mathbf{y}\|_2^2 + \beta \|\mathbf{w}^{(L)}\|_1$$

where $\Theta_s := \{\theta \in \Theta : \gamma_j^{(L-1)^2} + \alpha_j^{(L-1)^2} = 1, \forall j \in [m_{L-1}]\}$. $\qquad\square$

### B.3 DERIVATION OF THE DUAL PROBLEM IN (5)

Let us start with reparameterizing the scaled primal problem in (4) as

$$p_L^* = \min_{\theta \in \Theta_s, \hat{\mathbf{y}} \in \mathbb{R}^n} \frac{1}{2} \|\hat{\mathbf{y}} - \mathbf{y}\|_2^2 + \beta \left\| \mathbf{w}^{(L)} \right\|_1 \text{ s.t. } \hat{\mathbf{y}} = \sum_{j=1}^{m_{L-1}} \left( \mathrm{BN}_{\gamma,\alpha} \left( \mathbf{A}^{(L-2)}\mathbf{w}_j^{(L-1)} \right) \right)_+ w_j^{(L)}. \tag{23}$$

where $\Theta_s := \{\theta \in \Theta : \gamma_j^{(L-1)^2} + \alpha_j^{(L-1)^2} = 1, \forall j \in [m_{L-1}]\}$. We now form the following Lagrangian

$$L(\mathbf{v}, \hat{\mathbf{y}}, \mathbf{w}^{(L)}) = \frac{1}{2} \|\hat{\mathbf{y}} - \mathbf{y}\|_2^2 - \mathbf{v}^T \hat{\mathbf{y}} + \beta \left\| \mathbf{w}^{(L)} \right\|_1 + \mathbf{v}^T \sum_{j=1}^{m_{L-1}} \left( \mathrm{BN}_{\gamma,\alpha} \left( \mathbf{A}^{(L-2)}\mathbf{w}_j^{(L-1)} \right) \right)_+ w_j^{(L)}.$$

Then, the corresponding dual function is

$$g(\mathbf{v}) = \min_{\hat{\mathbf{y}}, \mathbf{w}^{(L)}} L(\mathbf{v}, \hat{\mathbf{y}}, \mathbf{w}^{(L)})$$

$$= -\frac{1}{2}\|\mathbf{v} - \mathbf{y}\|_2^2 + \frac{1}{2}\|\mathbf{y}\|_2^2 \ \text{s.t.} \ \left|\mathbf{v}^\top \left(\mathrm{BN}_{\gamma,\alpha}\left(\mathbf{A}^{(L-2)}\mathbf{w}_j^{(L-1)}\right)\right)_+\right| \leq \beta, \forall j \in [m_{L-1}].$$

Hence, the dual of (23) with respect to $\hat{\mathbf{y}}$ and $\mathbf{w}^{(L)}$ is

$$p_L^* = \min_{\theta \in \Theta_s} \max_{\mathbf{v}} g(\mathbf{v}) = \min_{\theta \in \Theta_s} \max_{\mathbf{v}} -\frac{1}{2}\|\mathbf{v} - \mathbf{y}\|_2^2 + \frac{1}{2}\|\mathbf{y}\|_2^2 \ \text{s.t.} \ \left|\mathbf{v}^\top \left(\mathrm{BN}_{\gamma,\alpha}\left(\mathbf{A}^{(L-2)}\mathbf{w}_j^{(L-1)}\right)\right)_+\right| \leq \beta, \forall j \in [m_{L-1}].$$

We then change the order of min-max to obtain the following lower bound

$$p_L^* \geq d_L^* := \max_{\mathbf{v}} -\frac{1}{2}\|\mathbf{v} - \mathbf{y}\|_2^2 + \frac{1}{2}\|\mathbf{y}\|_2^2 \ \text{s.t.} \ \max_{\theta \in \Theta_s} \left|\mathbf{v}^\top \left(\mathrm{BN}_{\gamma,\alpha}\left(\mathbf{A}^{(L-2)}\mathbf{w}^{(L-1)}\right)\right)_+\right| \leq \beta.$$

$\square$

### B.4 HYPERPLANE ARRANGEMENTS

We first define the set of all hyperplane arrangements for the data matrix $\mathbf{X}$ as

$$\mathcal{H} := \bigcup \left\{\{\mathrm{sign}(\mathbf{X}\mathbf{w})\} : \mathbf{w} \in \mathbb{R}^d\right\},$$

where $|\mathcal{H}| \leq 2^n$. We now define a new set to denote the indices with positive signs for each element in the set $\mathcal{H}$ as $\mathcal{S} := \left\{\{\cup_{h_i=1}\{i\}\} : \mathbf{h} \in \mathcal{H}\right\}$. With this definition, we note that given an element $S \in \mathcal{S}$, one can introduce a diagonal matrix $\mathbf{D}(S) \in \mathbb{R}^{n \times n}$ defined as $\mathbf{D}(S)_{ii} := \mathbb{1}[i \in S]$. Therefore, the output of ReLU activation can be equivalently written as $(\mathbf{X}\mathbf{w})_+ = \mathbf{D}(S)\mathbf{X}\mathbf{w}$ provided that $\mathbf{D}(S)\mathbf{X}\mathbf{w} \geq 0$ and $(\mathbf{I}_n - \mathbf{D}(S))\mathbf{X}\mathbf{w} \leq 0$ are satisfied. One can define more compactly these two constraints as $(2\mathbf{D}(S) - \mathbf{I}_n)\mathbf{X}\mathbf{w} \geq 0$. We now denote the cardinality of $\mathcal{S}$ as $P$, and obtain the following upperbound

$$P \leq 2 \sum_{k=0}^{r-1} \binom{n-1}{k} \leq 2r \left(\frac{e(n-1)}{r}\right)^r$$

where $r := \mathrm{rank}(\mathbf{X}) \leq \min(n, d)$ (Ojha, 2000; Stanley et al., 2004; Winder, 1966; Cover, 1965). For the rest of the paper, we enumerate all possible hyperplane arrangements in an arbitrary order and denote them as $\{\mathbf{D}_i\}_{i=1}^P$.

## C TWO-LAYER NETWORKS

### C.1 CLOSED-FORM SOLUTION TO THE DUAL PROBLEM IN (7)

**Lemma C.1.** *Suppose $n \leq d$ and $\mathbf{X}$ is full row-rank, then an optimal solution to (7) is*

$$\mathbf{v}^* = \begin{cases} \beta \frac{(\mathbf{y})_+}{\|(\mathbf{y})_+\|_2} - \beta \frac{(-\mathbf{y})_+}{\|(-\mathbf{y})_+\|_2} & \text{if } \beta \leq \|(\mathbf{y})_+\|_2, \beta \leq \|(-\mathbf{y})_+\|_2 \\ (\mathbf{y})_+ - \beta \frac{(-\mathbf{y})_+}{\|(-\mathbf{y})_+\|_2} & \text{if } \beta > \|(\mathbf{y})_+\|_2, \beta \leq (-\mathbf{y})_+\|_2 \\ \beta \frac{(\mathbf{y})_+}{\|(\mathbf{y})_+\|_2} - (-\mathbf{y})_+ & \text{if } \beta \leq \|(\mathbf{y})_+\|_2, \beta > \|(-\mathbf{y})_+\|_2 \\ \mathbf{y} & \text{if } \beta > \|(\mathbf{y})_+\|_2, \beta > \|(-\mathbf{y})_+\|_2 \end{cases}.$$

*Proof.* Since $n \leq d$ and $\mathbf{X}$ is full row-rank, we have

$$\max_{\theta \in \Theta_s} \left|\mathbf{v}^T \left(\frac{(\mathbf{I}_n - \frac{1}{n}\mathbf{1}\mathbf{1}^T)\mathbf{X}\mathbf{w}^{(1)}}{\|(\mathbf{I}_n - \frac{1}{n}\mathbf{1}\mathbf{1}^T)\mathbf{X}\mathbf{w}^{(1)}\|_2}\gamma^{(1)} + \frac{1}{\sqrt{n}}\alpha^{(1)}\right)_+\right| \leq \max\left\{\|(\mathbf{v})_+\|_2, \|(-\mathbf{v})_+\|_2\right\} \max_{\substack{\mathbf{1}^T\mathbf{u}=0 \\ \|\mathbf{u}\|_2=\|\mathbf{s}\|_2=1}} \left\|\left(\begin{bmatrix}\mathbf{u} & \frac{1}{\sqrt{n}}\end{bmatrix}\mathbf{s}\right)_+\right\|_2$$

$$\leq \max\left\{\|(\mathbf{v})_+\|_2, \|(-\mathbf{v})_+\|_2\right\} \max_{\|\mathbf{u}\|_2=1} \|(\mathbf{u})_+\|_2$$

$$\leq \max\left\{\|(\mathbf{v})_+\|_2, \|(-\mathbf{v})_+\|_2\right\},$$

where $\mathbf{s} = \begin{bmatrix} \gamma^{(1)} & \alpha^{(1)} \end{bmatrix}^T$ and the equality is achieved when

$$\mathbf{w}^{(1)} = \mathbf{X}^\dagger(\mathbf{v} - \min_i v_i), \ \mathbf{s} = \frac{1}{\|\mathbf{v}\|_2}\begin{bmatrix} \|\mathbf{v} - \frac{1}{n}\mathbf{1}\mathbf{1}^T\mathbf{v}\|_2 \\ \frac{1}{\sqrt{n}}\mathbf{1}^T\mathbf{v} \end{bmatrix}.$$

Thus, the dual problem (7) can be reformulated as

$$\max_{\mathbf{v}} -\frac{1}{2}\|\mathbf{v} - \mathbf{y}\|_2^2 + \frac{1}{2}\|\mathbf{y}\|_2^2 \ \text{s.t.} \ \max\left\{\|(\mathbf{v})_+\|_2, \|(-\mathbf{v})_+\|_2\right\} \leq \beta.$$

The problem above can be maximized using the following optimal dual parameter

$$\mathbf{v}^* = \begin{cases} \beta\frac{(\mathbf{y})_+}{\|(\mathbf{y})_+\|_2} - \beta\frac{(-\mathbf{y})_+}{\|(-\mathbf{y})_+\|_2} & \text{if } \beta \leq \|(\mathbf{y})_+\|_2, \beta \leq \|(-\mathbf{y})_+\|_2 \\ (\mathbf{y})_+ - \beta\frac{(-\mathbf{y})_+}{\|(-\mathbf{y})_+\|_2} & \text{if } \beta > \|(\mathbf{y})_+\|_2, \beta \leq (-\mathbf{y})_+\|_2 \\ \beta\frac{(\mathbf{y})_+}{\|(\mathbf{y})_+\|_2} - (-\mathbf{y})_+ & \text{if } \beta \leq \|(\mathbf{y})_+\|_2, \beta > \|(-\mathbf{y})_+\|_2 \\ \mathbf{y} & \text{if } \beta > \|(\mathbf{y})_+\|_2, \beta > \|(-\mathbf{y})_+\|_2 \end{cases}.$$

$\square$

## C.2 PROOF OF THEOREM 2.1

For the solution in Lemma C.1, the corresponding extreme points for the two-sided constraint are

$$\mathbf{W}^{(1)*} = \begin{cases} \left(\mathbf{X}^\dagger(\mathbf{y})_+, \mathbf{X}^\dagger(-\mathbf{y})_+\right) & \text{if } \beta \leq \|(\mathbf{y})_+\|_2, \beta \leq \|(-\mathbf{y})_+\|_2 \\ \mathbf{X}^\dagger(-\mathbf{y})_+ & \text{if } \beta > \|(\mathbf{y})_+\|_2, \beta \leq (-\mathbf{y})_+\|_2 \\ \mathbf{X}^\dagger(\mathbf{y})_+ & \text{if } \beta \leq \|(\mathbf{y})_+\|_2, \beta > \|(\mathbf{y})_+\|_2 \\ \mathbf{0} & \text{if } \beta > \|(\mathbf{y})_+\|_2, \beta > \|(-\mathbf{y})_+\|_2 \end{cases}$$

and

$$\begin{bmatrix} \gamma_1^{(1)*} \\ \alpha_1^{(1)*} \end{bmatrix} = \frac{1}{\|(\mathbf{y})_+\|_2}\begin{bmatrix} \|(\mathbf{y})_+ - \frac{1}{n}\mathbf{1}\mathbf{1}^T(\mathbf{y})_+\|_2 \\ \frac{1}{\sqrt{n}}\mathbf{1}^T(\mathbf{y})_+ \end{bmatrix} \quad \begin{bmatrix} \gamma_2^{(1)*} \\ \alpha_2^{(1)*} \end{bmatrix} = \frac{1}{\|(-\mathbf{y})_+\|_2}\begin{bmatrix} \|(-\mathbf{y})_+ - \frac{1}{n}\mathbf{1}\mathbf{1}^T(-\mathbf{y})_+\|_2 \\ \frac{1}{\sqrt{n}}\mathbf{1}^T(-\mathbf{y})_+ \end{bmatrix}.$$

We now substitute these solutions into the primal problem (6) and then take the derivative with respect to the output layer weights $\mathbf{w}^{(2)}$. Since the primal problem is linear and convex provided that the first layer weights are fixed as $\mathbf{W}^{(1)*}$, we obtain the following closed-form solutions for the output layer weights

$$\mathbf{w}^{(2)*} = \begin{cases} \begin{bmatrix} \|(\mathbf{y})_+\|_2 - \beta \\ -\|(-\mathbf{y})_+\|_2 + \beta \end{bmatrix} & \text{if } \beta \leq \|(\mathbf{y})_+\|_2, \beta \leq \|(-\mathbf{y})_+\|_2 \\ -\|(-\mathbf{y})_+\|_2 + \beta & \text{if } \beta > \|(\mathbf{y})_+\|_2, \beta \leq \|(-\mathbf{y})_+\|_2 \\ \|(\mathbf{y})_+\|_2 - \beta & \text{if } \beta \leq \|(\mathbf{y})_+\|_2, \beta > \|(-\mathbf{y})_+\|_2 \\ 0 & \text{if } \beta > \|(\mathbf{y})_+\|_2, \beta > \|(-\mathbf{y})_+\|_2 \end{cases}.$$

These set of weights, $\{\mathbf{W}^{(1)*}, \gamma^{(1)*}, \alpha^{(1)*}, \mathbf{w}^{(2)*}\}$, achieves $p_2^* = d_2^*$, therefore, strong duality holds.

$\square$

## C.3 PROOF OF THEOREM 2.2

We first note that if the number of neurons satisfies $m_1 \geq m^*$ for some $m^* \in \mathbb{N}$, where $m^* \leq n + 1$, then strong duality holds, i.e., $p_2^* = d_2^*$ (see Section 2 of Pilanci & Ergen (2020)). In the weakly regularized regime where the network interpolates the training data, there are even simpler ways to verify this upperbound, e.g., Zhang et al. (2021) show how to construct a solution that exactly interpolates the training data with $m_1 = n$, and therefore the ReLU network fits any real valued $\mathbf{y} \in \mathbb{R}^n$ with $n$ neurons.

Following the steps in Pilanci & Ergen (2020), we first focus on the dual constraint in (7)

$$\max_{\substack{\mathbf{w}^{(1)} \\ \gamma^{(1)2}+\alpha^{(1)2}=1}} \left| \mathbf{v}^T \left( \frac{(\mathbf{I}_n - \frac{1}{n}\mathbf{1}\mathbf{1}^T)\mathbf{X}\mathbf{w}^{(1)}}{\|(\mathbf{I}_n - \frac{1}{n}\mathbf{1}\mathbf{1}^T)\mathbf{X}\mathbf{w}^{(1)}\|_2}\gamma^{(1)} + \frac{1}{\sqrt{n}}\alpha^{(1)} \right)_+ \right| = \max_{\|\mathbf{q}\|_2=\|\mathbf{s}\|_2=1} \left| \mathbf{v}^T \left( \begin{bmatrix} \mathbf{U}\mathbf{q} & \frac{1}{\sqrt{n}} \end{bmatrix} \mathbf{s} \right)_+ \right|$$

$$= \max_{\|\mathbf{s}'\|_2=1} \left| \mathbf{v}^T \left( \mathbf{U}'\mathbf{s}' \right)_+ \right|$$

$$= \max_{i\in[P]} \max_{\substack{\|\mathbf{s}'\|_2=1 \\ (2\mathbf{D}_i-\mathbf{I}_n)\mathbf{U}'\mathbf{s}'\geq 0}} \left| \mathbf{v}^T \mathbf{D}_i \mathbf{U}'\mathbf{s}' \right|,$$

where $\mathbf{q} := \mathbf{\Sigma}\mathbf{V}^T\mathbf{w}^{(1)}/\|\mathbf{\Sigma}\mathbf{V}^T\mathbf{w}^{(1)}\|_2$, $\mathbf{s} := \begin{bmatrix} \gamma^{(1)} & \alpha^{(1)} \end{bmatrix}^T$, $\mathbf{s}' := \begin{bmatrix} \gamma^{(1)}\mathbf{q}^T & \alpha^{(1)} \end{bmatrix}^T$, and $\mathbf{U}' := \begin{bmatrix} \mathbf{U} & \frac{1}{\sqrt{n}} \end{bmatrix}$. Now, let us consider the single side of this constraint and introduce a dual variable $\boldsymbol{\rho}$ for the constraint as follows

$$\max_{i\in[P]} \max_{\substack{\|\mathbf{s}'\|_2=1 \\ (2\mathbf{D}_i-\mathbf{I}_n)\mathbf{U}'\mathbf{s}'\geq 0}} \mathbf{v}^T\mathbf{D}_i\mathbf{U}'\mathbf{s}' = \max_{i\in[P]} \min_{\boldsymbol{\rho}\geq 0} \max_{\|\mathbf{s}'\|_2=1} \left( \mathbf{v}^T\mathbf{D}_i\mathbf{U}' + \boldsymbol{\rho}^T(2\mathbf{D}_i-\mathbf{I}_n)\mathbf{U}' \right)\mathbf{s}'$$

$$= \max_{i\in[P]} \min_{\boldsymbol{\rho}\geq 0} \left\| \mathbf{v}^T\mathbf{D}_i\mathbf{U}' + \boldsymbol{\rho}^T(2\mathbf{D}_i-\mathbf{I}_n)\mathbf{U}' \right\|_2.$$

Therefore, we have

$$\max_{\theta\in\Theta_s} \mathbf{v}^T \left( \frac{(\mathbf{I}_n - \frac{1}{n}\mathbf{1}\mathbf{1}^T)\mathbf{X}\mathbf{w}^{(1)}}{\|(\mathbf{I}_n - \frac{1}{n}\mathbf{1}\mathbf{1}^T)\mathbf{X}\mathbf{w}^{(1)}\|_2}\gamma^{(1)} + \frac{1}{\sqrt{n}}\alpha^{(1)} \right)_+ \leq \beta \Longleftrightarrow \forall i\in[P], \min_{\boldsymbol{\rho}\geq 0} \left\| \mathbf{v}^T\mathbf{D}_i\mathbf{U}' + \boldsymbol{\rho}^T(2\mathbf{D}_i-\mathbf{I}_n)\mathbf{U}' \right\|_2 \leq \beta$$

$$\Longleftrightarrow \forall i\in[P], \exists\boldsymbol{\rho}_i\geq 0 \text{ s.t. } \left\| \mathbf{v}^T\mathbf{D}_i\mathbf{U}' + \boldsymbol{\rho}_i^T(2\mathbf{D}_i-\mathbf{I}_n)\mathbf{U}' \right\|_2 \leq \beta.$$

Similar arguments also hold for the negative side of the absolute value constraint. Therefore, we can reformulate (7) as a convex optimization problem with variables $\mathbf{v}, \boldsymbol{\rho}_i, \boldsymbol{\rho}'_i \in \mathbb{R}^n, \forall i\in[P]$ and $2P$ constraints as follows

$$d_2^* = \max_{\mathbf{v},\boldsymbol{\rho}_i} -\frac{1}{2}\|\mathbf{v}-\mathbf{y}\|_2^2 + \frac{1}{2}\|\mathbf{y}\|_2^2 \text{ s.t. } \begin{array}{l} \left\| \mathbf{v}^T\mathbf{D}_i\mathbf{U}' + \boldsymbol{\rho}_i^T(2\mathbf{D}_i-\mathbf{I}_n)\mathbf{U}' \right\|_2 \leq \beta \\ \left\| -\mathbf{v}^T\mathbf{D}_i\mathbf{U}' + \boldsymbol{\rho}_i'^T(2\mathbf{D}_i-\mathbf{I}_n)\mathbf{U}' \right\|_2 \leq \beta \end{array}, \forall i\in[P]. \quad (24)$$

As noted in Pilanci & Ergen (2020), since the problem in (24) is strictly feasible for the particular choice of parameters $\mathbf{v} = \boldsymbol{\rho}_i = \boldsymbol{\rho}'_i = \mathbf{0}, \forall\in[P]$, Slater's condition and therefore strong duality holds (Boyd & Vandenberghe, 2004). We now write (24) in a Lagrangian form with the parameters $\boldsymbol{\lambda}, \boldsymbol{\lambda}' \in \mathbb{R}^P$ as

$$d_2^* = \min_{\boldsymbol{\lambda},\boldsymbol{\lambda}'\geq 0} \max_{\substack{\mathbf{v} \\ \boldsymbol{\rho}_i\geq 0,\forall i}} -\frac{1}{2}\|\mathbf{v}-\mathbf{y}\|_2^2 + \frac{1}{2}\|\mathbf{y}\|_2^2 + \sum_{i=1}^P \lambda_i(\beta - \left\| \mathbf{v}^T\mathbf{D}_i\mathbf{U}' + \boldsymbol{\rho}_i^T(2\mathbf{D}_i-\mathbf{I}_n)\mathbf{U}' \right\|_2)$$

$$+ \sum_{i=1}^P \lambda'_i(\beta - \left\| -\mathbf{v}^T\mathbf{D}_i\mathbf{U}' + \boldsymbol{\rho}_i'^T(2\mathbf{D}_i-\mathbf{I}_n)\mathbf{U}' \right\|_2). \quad (25)$$

We next introduce additional variables $\mathbf{r}_i, \mathbf{r}'_i \in \mathbb{R}^d, \forall i\in[P]$ to represent (25) as

$$d_2^* = \min_{\boldsymbol{\lambda},\boldsymbol{\lambda}'\geq 0} \max_{\substack{\mathbf{v} \\ \boldsymbol{\rho}_i\geq 0,\forall i}} \min_{\substack{\mathbf{r}_i:\|\mathbf{r}_i\|_2\leq 1 \\ \mathbf{r}'_i\|\mathbf{r}'_i\|_2\leq 1}} -\frac{1}{2}\|\mathbf{v}-\mathbf{y}\|_2^2 + \frac{1}{2}\|\mathbf{y}\|_2^2 + \sum_{i=1}^P \lambda_i \left( \beta - \left( \mathbf{v}^T\mathbf{D}_i\mathbf{U}' + \boldsymbol{\rho}_i^T(2\mathbf{D}_i-\mathbf{I}_n)\mathbf{U}' \right)\mathbf{r}_i \right)$$

$$+ \sum_{i=1}^P \lambda'_i \left( \beta + \left( -\mathbf{v}^T\mathbf{D}_i\mathbf{U}' + \boldsymbol{\rho}_i'^T(2\mathbf{D}_i-\mathbf{I}_n)\mathbf{U}' \right)\mathbf{r}'_i \right). \quad (26)$$

Now, we remark that the objective function in (26) is concave in $\mathbf{v}, \boldsymbol{\rho}_i, \boldsymbol{\rho}'_i$ and convex in $\mathbf{r}_i, \mathbf{r}'_i, \forall i\in[P]$. In addition, the constraint set for the additional variables, i.e., $\|\mathbf{r}_i\|_2, \|\mathbf{r}'_i\|_2 \leq 1$, are convex and

compact. Then, we use Sion's minimax theorem (Sion, 1958) to change the order of the inner max-min problems as

$$
d_2^* = \min_{\substack{\boldsymbol{\lambda}, \boldsymbol{\lambda}' \geq 0 \\ }} \min_{\substack{\mathbf{r}_i: \|\mathbf{r}_i\|_2 \leq 1 \\ \mathbf{r}_i' \|\mathbf{r}_i'\|_2 \leq 1}} \max_{\substack{\mathbf{v} \\ \boldsymbol{\rho}_i \geq 0, \forall i}} -\frac{1}{2}\|\mathbf{v} - \mathbf{y}\|_2^2 + \frac{1}{2}\|\mathbf{y}\|_2^2 + \sum_{i=1}^{P} \lambda_i \left( \beta - \left( \mathbf{v}^T \mathbf{D}_i \mathbf{U}' + \boldsymbol{\rho}_i^T (2\mathbf{D}_i - \mathbf{I}_n)\mathbf{U}' \right) \mathbf{r}_i \right)
$$
$$
+ \sum_{i=1}^{P} \lambda_i' \left( \beta + \left( -\mathbf{v}^T \mathbf{D}_i \mathbf{U}' + \boldsymbol{\rho}_i'^T (2\mathbf{D}_i - \mathbf{I}_n)\mathbf{U}' \right) \mathbf{r}_i' \right). \tag{27}
$$

Thus, we can now perform the maximization over the variables $\mathbf{v}, \boldsymbol{\rho}_i, \boldsymbol{\rho}_i'$ to simplify (27) into the following form

$$
p_{2c}^* = \min_{\substack{\boldsymbol{\lambda}, \boldsymbol{\lambda}' \geq 0 \\ }} \min_{\substack{\mathbf{r}_i: \|\mathbf{r}_i\|_2 \leq 1 \\ \mathbf{r}_i' \|\mathbf{r}_i'\|_2 \leq 1}} \frac{1}{2} \left\| \sum_{i=1}^{P} (\lambda_i \mathbf{D}_i \mathbf{U}' \mathbf{r}_i - \lambda_i' \mathbf{D}_i \mathbf{U}' \mathbf{r}_i') - \mathbf{y} \right\|_2^2 + \beta \sum_{i=1}^{P} (\lambda_i + \lambda_i') \tag{28}
$$
$$
\text{s.t. } (2\mathbf{D}_i - \mathbf{I}_n)\mathbf{U}' \mathbf{r}_i \geq 0, \ (2\mathbf{D}_i - \mathbf{I}_n)\mathbf{U}' \mathbf{r}_i' \geq 0, \ \forall i \in [P].
$$

We then apply a set variable changes as $\mathbf{s}_i = \lambda_i \mathbf{r}_i$ and $\mathbf{s}_i' = \lambda_i' \mathbf{r}_i'$, $\forall i \in [P]$ to obtain the following problem

$$
p_{2c}^* = \min_{\substack{\boldsymbol{\lambda}, \boldsymbol{\lambda}' \geq 0 \\ }} \min_{\substack{\mathbf{s}_i: \|\mathbf{s}_i\|_2 \leq \lambda_i \\ \mathbf{s}_i' \|\mathbf{s}_i'\|_2 \leq \lambda_i'}} \frac{1}{2} \left\| \sum_{i=1}^{P} \mathbf{D}_i \mathbf{U}' (\mathbf{s}_i - \mathbf{s}_i') - \mathbf{y} \right\|_2^2 + \beta \sum_{i=1}^{P} (\lambda_i + \lambda_i') \tag{29}
$$
$$
\text{s.t. } (2\mathbf{D}_i - \mathbf{I}_n)\mathbf{U}' \mathbf{s}_i \geq 0, \ (2\mathbf{D}_i - \mathbf{I}_n)\mathbf{U}' \mathbf{s}_i' \geq 0, \ \forall i \in [P].
$$

Finally, since the optimality conditions require $\|\mathbf{s}_i\|_2 = \lambda_i$ and $\|\mathbf{s}_i'\|_2 = \lambda_i'$, $\forall i \in [P]$, we write (29) as a single convex minimization problem as

$$
p_{2c}^* = \min_{\mathbf{s}_i, \mathbf{s}_i'} \frac{1}{2} \left\| \sum_{i=1}^{P} \mathbf{D}_i \mathbf{U}' (\mathbf{s}_i - \mathbf{s}_i') - \mathbf{y} \right\|_2^2 + \beta \sum_{i=1}^{P} (\|\mathbf{s}_i\|_2 + \|\mathbf{s}_i'\|_2) \tag{30}
$$
$$
\text{s.t. } (2\mathbf{D}_i - \mathbf{I}_n)\mathbf{U}' \mathbf{s}_i \geq 0, \ (2\mathbf{D}_i - \mathbf{I}_n)\mathbf{U}' \mathbf{s}_i' \geq 0, \ \forall i \in [P].
$$

Now, we note that since (24) is convex and Slater's condition holds as shown above, we have $p_{2c}^* = d_2^*$. In addition, as proven in Pilanci & Ergen (2020) strong duality also holds for the original non-convex problem, i.e., $p_2^* = d_2^*$, therefore, we get $p_2^* = p_{2c}^* = d_2^*$.

**Constructing an optimal solution to the primal problem:**
One can also construct an optimal solution to the non-convex training problem in (6) as follows

$$
\left( \mathbf{w}_{j_{1i}}^{(1)^*}, \gamma_{j_{1i}}^{(1)^*}, \alpha_{j_{1i}}^{(1)^*}, w_{j_{1i}}^{(2)^*} \right) = \left( \mathbf{V}\boldsymbol{\Sigma}^{-1} \mathbf{s}_{i,1:r}^*, \frac{\|\mathbf{s}_{i,1:r}^*\|_2}{\|\mathbf{s}_i^*\|_2^{\frac{1}{2}}}, \frac{s_{i,r+1}^*}{\|\mathbf{s}_i^*\|_2^{\frac{1}{2}}}, \|\mathbf{s}_i^*\|_2^{\frac{1}{2}} \right), \text{ if } \mathbf{s}_i^* \neq 0
$$

$$
\left( \mathbf{w}_{j_{2i}}^{(1)^*}, \gamma_{j_{2i}}^{(1)^*}, \alpha_{j_{2i}}^{(1)^*}, w_{j_{2i}}^{(2)^*} \right) = \left( \mathbf{V}\boldsymbol{\Sigma}^{-1} \mathbf{s}_{i,1:r}'^*, \frac{\|\mathbf{s}_{i,1:r}'^*\|_2}{\|\mathbf{s}_i'^*\|_2^{\frac{1}{2}}}, \frac{s_{i,r+1}'^*}{\|\mathbf{s}_i'^*\|_2^{\frac{1}{2}}}, -\|\mathbf{s}_i'^*\|_2^{\frac{1}{2}} \right), \text{ if } \mathbf{s}_i'^* \neq 0,
$$

where $\{\mathbf{s}_i^*, \mathbf{s}_i'^*\}_{i=1}^P$ are the optimal solutions to (8) and and $j_{si} \in [|\mathcal{J}_s|]$ given the definitions $\mathcal{J}_1 := \{i : \mathbf{s}_i^* \neq 0\}$ and $\mathcal{J}_2 := \{i : \mathbf{s}_i'^* \neq 0\}$. We also note that here the second subscript denotes the selected entries of the corresponding vector, e.g., $\mathbf{s}_{i,1:r}$ denotes the first $r$ entries of the vector $\mathbf{s}_i$. Thus, we have $m^* = |\mathcal{J}_1| + |\mathcal{J}_2|$.

**Verifying strong duality:**
Next, we verify strong duality, i.e., $p_2^* = p_{2c}^* = d_2^*$, by plugging the optimal solution to the non-convex objective in (3) and the corresponding convex objective in (30).

Given the optimal solutions $\{\mathbf{s}_i^*, \mathbf{s}_i'^*\}_{i=1}^P$ to (30), the convex problem has the following objective

$$
p_{2c}^* = \frac{1}{2} \left\| \sum_{i=1}^{P} \mathbf{D}_i \mathbf{U}' (\mathbf{s}_i^* - \mathbf{s}_i'^*) - \mathbf{y} \right\|_2^2 + \beta \sum_{i=1}^{P} (\|\mathbf{s}_i^*\|_2 + \|\mathbf{s}_i'^*\|_2) \tag{31}
$$

and the corresponding non-convex problem objective is as follows

$$p_2^* = \frac{1}{2} \|f_{\theta,2}(\mathbf{X}) - \mathbf{y}\|_2^2 + \frac{\beta}{2} \sum_{j=1}^{m_1} \left( \gamma_j^{(1)*^2} + \alpha_j^{(1)*^2} + w_j^{(2)*^2} \right). \tag{32}$$

We first prove that the first terms in both objectives have the same value. To do that, we compute the output of the non-convex neural network with the optimal weight construction above

$$
\begin{aligned}
f_{\theta,2}(\mathbf{X}) &= \sum_{j=1}^{m_1} \left( \frac{(\mathbf{I}_n - \frac{1}{n}\mathbf{11}^T)\mathbf{X}\mathbf{w}_j^{(1)*}}{\|(\mathbf{I}_n - \frac{1}{n}\mathbf{11}^T)\mathbf{X}\mathbf{w}_j^{(1)*}\|_2} \gamma_j^{(1)*} + \frac{1}{\sqrt{n}}\alpha_j^{(1)*} \right)_+ w_j^{(2)*} \\
&= \sum_{i \in \mathcal{J}_1} \left( \frac{\mathbf{U}\mathbf{s}_{i,1:r}^*}{\|\mathbf{U}\mathbf{s}_{i,1:r}^*\|_2} \frac{\|\mathbf{s}_{i,1:r}^*\|_2}{\|\mathbf{s}_i^*\|_2^{\frac{1}{2}}} + \frac{1}{\sqrt{n}} \frac{s_{i,r+1}^*}{\|\mathbf{s}_i^*\|_2^{\frac{1}{2}}} \right)_+ \|\mathbf{s}_i^*\|_2^{\frac{1}{2}} \\
&\quad - \sum_{i \in \mathcal{J}_2} \left( \frac{\mathbf{U}\mathbf{s}_{i,1:r}'^*}{\|\mathbf{U}\mathbf{s}_{i,1:r}'^*\|_2} \frac{\|\mathbf{s}_{i,1:r}'^*\|_2}{\|\mathbf{s}_i'^*\|_2^{\frac{1}{2}}} + \frac{1}{\sqrt{n}} \frac{s_{i,r+1}'^*}{\|\mathbf{s}_i'^*\|_2^{\frac{1}{2}}} \right)_+ \|\mathbf{s}_i'^*\|_2^{\frac{1}{2}} \\
&= \sum_{i \in \mathcal{J}_1} (\mathbf{U}'\mathbf{s}_i^*)_+ - \sum_{i \in \mathcal{J}_2} \left( \mathbf{U}'\mathbf{s}_i'^* \right)_+ \\
&= \sum_{i=1}^{P} \mathbf{D}_i \mathbf{U}'(\mathbf{s}_i^* - \mathbf{s}_i'^*), \tag{33}
\end{aligned}
$$

where the last equality follows from the constraints in (30).

Next, we prove that the regularizations terms have the same objective as follows

$$
\begin{aligned}
\frac{\beta}{2} \sum_{j=1}^{m_1} \left( \gamma_j^{(1)*^2} + \alpha_j^{(1)*^2} + w_j^{(2)*^2} \right) &= \frac{\beta}{2} \sum_{j \in \mathcal{J}_1} \left( \left( \frac{\|\mathbf{s}_{i,1:r}^*\|_2}{\|\mathbf{s}_i^*\|_2^{\frac{1}{2}}} \right)^2 + \left( \frac{s_{i,r+1}^*}{\|\mathbf{s}_i^*\|_2^{\frac{1}{2}}} \right)^2 + \left( \|\mathbf{s}_i^*\|_2^{\frac{1}{2}} \right)^2 \right) \\
&\quad + \frac{\beta}{2} \sum_{i \in \mathcal{J}_2} \left( \left( \frac{\|\mathbf{s}_{i,1:r}'^*\|_2}{\|\mathbf{s}_i'^*\|_2^{\frac{1}{2}}} \right)^2 + \left( \frac{s_{i,r+1}'^*}{\|\mathbf{s}_i'^*\|_2^{\frac{1}{2}}} \right)^2 + \left( -\|\mathbf{s}_i'^*\|_2^{\frac{1}{2}} \right)^2 \right) \\
&= \frac{\beta}{2} \sum_{i \in \mathcal{J}_1} \left( \frac{\|\mathbf{s}_{i,1:r}^*\|_2^2}{\|\mathbf{s}_i^*\|_2} + \frac{s_{i,r+1}^{*^2}}{\|\mathbf{s}_i^*\|_2} + \|\mathbf{s}_i^*\|_2 \right) \\
&\quad + \frac{\beta}{2} \sum_{i \in \mathcal{J}_2} \left( \frac{\|\mathbf{s}_{i,1:r}'^*\|_2^2}{\|\mathbf{s}_i'^*\|_2} + \frac{s_{i,r+1}'^{*^2}}{\|\mathbf{s}_i'^*\|_2} + \|\mathbf{s}_i'^*\|_2 \right) \\
&= \beta \sum_{i \in \mathcal{J}_1} \|\mathbf{s}_i^*\|_2 + \beta \sum_{i \in \mathcal{J}_2} \|\mathbf{s}_i'^*\|_2 \\
&= \beta \sum_{i=1}^{P} (\|\mathbf{s}_i^*\|_2 + \|\mathbf{s}_i'^*\|_2). \tag{34}
\end{aligned}
$$

Therefore, by (33) and (34), the non-convex objective (32) and the convex objective (31) are the same, i.e., $p_2^* = p_{2c}^*$. $\qquad\square$

## C.4   PROOF OF LEMMA 2.1

We note that similar approaches are also presented in Sahiner et al. (2021a); **?**.

For any $\theta \in \Theta$, we can rescale the parameters as $\bar{\gamma}_j^{(1)} = \eta_j \gamma_j^{(1)}$, $\bar{\alpha}_j^{(1)} = \eta_j \alpha_j^{(1)}$ and $\bar{\mathbf{w}}_j^{(2)} = \mathbf{w}_j^{(2)}/\eta_j$, for any $\eta_j > 0$. Then, the network output becomes

$$
\begin{aligned}
f_{\bar{\theta},2}(\mathbf{X}) &= \sum_{j=1}^{m_1} \left( \frac{(\mathbf{I}_n - \frac{1}{n}\mathbf{1}\mathbf{1}^T)\mathbf{X}\mathbf{w}_j^{(1)}}{\|(\mathbf{I}_n - \frac{1}{n}\mathbf{1}\mathbf{1}^T)\mathbf{X}\mathbf{w}_j^{(1)}\|_2}\bar{\gamma}_j^{(1)} + \mathbf{1}\bar{\alpha}_j^{(1)} \right)_+ \bar{\mathbf{w}}_j^{(2)^T} \\
&= \sum_{j=1}^{m_1} \left( \frac{(\mathbf{I}_n - \frac{1}{n}\mathbf{1}\mathbf{1}^T)\mathbf{X}\mathbf{w}_j^{(1)}}{\|(\mathbf{I}_n - \frac{1}{n}\mathbf{1}\mathbf{1}^T)\mathbf{X}\mathbf{w}_j^{(1)}\|_2}\gamma_j^{(1)} + \mathbf{1}\alpha_j^{(1)} \right)_+ \mathbf{w}_j^{(2)^T} \\
&= f_{\theta,2}(\mathbf{X}),
\end{aligned}
$$

which proves $f_{\theta,2}(\mathbf{X}) = f_{\bar{\theta},2}(\mathbf{X})$. Moreover, we use the following inequality

$$
\frac{1}{2}\sum_{j=1}^{m_1} \left( \gamma_j^{(1)^2} + \alpha_j^{(1)^2} + \left\|\mathbf{w}_j^{(2)}\right\|_2^2 \right) \geq \sum_{j=1}^{m_1} \left\|\mathbf{w}_j^{(2)}\right\|_2 \sqrt{\gamma_j^{(1)^2} + \alpha_j^{(1)^2}},
$$

where the equality is achieved when

$$
\eta_j = \left( \frac{\left\|\mathbf{w}_j^{(2)}\right\|_2}{\sqrt{\gamma_j^{(1)^2} + \alpha_j^{(1)^2}}} \right)^{\frac{1}{2}}
$$

is used. Since this scaling does not alter the right-hand side of the inequality, without loss of generality, we can set $\sqrt{\gamma_j^{(1)^2} + \alpha_j^{(1)^2}} = 1, \forall j \in [m_1]$. Therefore, the right-hand side becomes $\sum_{j=1}^{m_1} \|\mathbf{w}_j^{(2)}\|_2$.

We then note that due to the arguments in Appendix B.2.1, we can remove the regularization on $\mathbf{w}_j^{(1)}$ without loss of generality. Therefore, the final optimization problem is as follows

$$
p_{2v}^* = \min_{\theta \in \Theta_s} \frac{1}{2}\|f_{\theta,2}(\mathbf{X}) - \mathbf{Y}\|_F^2 + \beta \sum_{j=1}^{m_1} \left\|\mathbf{w}_j^{(2)}\right\|_2,
$$

where $\Theta_s := \{\theta \in \Theta : \gamma_j^{(1)^2} + \alpha_j^{(1)^2} = 1, \forall j \in [m_1]\}$. $\qquad \square$

## C.5 Proof of Theorem 2.3

We first note that if the number of neurons satisfies $m_1 \geq m^*$ for some $m^* \in \mathbb{N}$, where $m^* \leq nC + 1$, then strong duality holds, i.e., $p_{2v}^* = d_{2v}^*$, (see Section A.3.1 of Sahiner et al. (2021a) for details). Following the steps in Sahiner et al. (2021a) and Theorem 2.2, we again focus on the dual constraint in (11)

$$
\begin{aligned}
\max_{\substack{\mathbf{w}^{(1)} \\ \gamma^{(1)^2}+\alpha^{(1)^2}=1}} & \left\|\mathbf{V}^T\left( \frac{(\mathbf{I}_n - \frac{1}{n}\mathbf{1}\mathbf{1}^T)\mathbf{X}\mathbf{w}^{(1)}}{\|(\mathbf{I}_n - \frac{1}{n}\mathbf{1}\mathbf{1}^T)\mathbf{X}\mathbf{w}^{(1)}\|_2}\gamma^{(1)} + \frac{\mathbf{1}}{\sqrt{n}}\alpha^{(1)} \right)_+\right\|_2 = \max_{\|\mathbf{q}\|_2=\|\mathbf{s}\|_2=1} \left\|\mathbf{V}^T\left( \begin{bmatrix} \mathbf{U}\mathbf{q} & \frac{1}{\sqrt{n}} \end{bmatrix}\mathbf{s} \right)_+\right\|_2 \\
& = \max_{\|\mathbf{s}'\|_2=1} \left\|\mathbf{V}^T\left(\mathbf{U}'\mathbf{s}'\right)_+\right\|_2 \\
& = \max_{\|\mathbf{g}\|_2=\|\mathbf{s}'\|_2=1} \mathbf{g}^T\mathbf{V}^T\left(\mathbf{U}'\mathbf{s}'\right)_+ \\
& = \max_{i \in [P]} \max_{\substack{\|\mathbf{g}\|_2=\|\mathbf{s}'\|_2=1 \\ (2\mathbf{D}_i-\mathbf{I}_n)\mathbf{U}'\mathbf{s}'\geq 0}} \mathbf{g}^T\mathbf{V}^T\mathbf{D}_i\mathbf{U}'\mathbf{s}' \\
& = \max_{i \in [P]} \max_{\substack{\|\mathbf{g}\|_2=\|\mathbf{s}'\|_2=1 \\ (2\mathbf{D}_i-\mathbf{I}_n)\mathbf{U}'\mathbf{s}'\geq 0}} \left\langle \mathbf{V}, \mathbf{D}_i\mathbf{U}'\mathbf{s}'\mathbf{g}^T \right\rangle \\
& = \max_{i \in [P]} \max_{\mathbf{S} \in \mathcal{K}_i} \left\langle \mathbf{V}, \mathbf{D}_i\mathbf{U}'\mathbf{S} \right\rangle,
\end{aligned}
$$

where $\mathbf{s}' = \begin{bmatrix} \gamma \mathbf{q}^T & \alpha \end{bmatrix}^T$, $\mathbf{U}' = \begin{bmatrix} \mathbf{U} & \frac{1}{\sqrt{n}} \end{bmatrix}$, and $\mathcal{K}_i := \mathrm{conv}\{\mathbf{Z} = \mathbf{s}' \mathbf{g}^T : (2\mathbf{D}_i - \mathbf{I}_n)\mathbf{U}'\mathbf{s}' \geq 0, \ \|\mathbf{Z}\|_* \leq 1\}$. Thus, we have

$$\max_{\theta \in \Theta_s} \left\| \mathbf{V}^T \left( \frac{(\mathbf{I}_n - \frac{1}{n}\mathbf{1}\mathbf{1}^T)\mathbf{X}\mathbf{w}^{(1)}}{\|(\mathbf{I}_n - \frac{1}{n}\mathbf{1}\mathbf{1}^T)\mathbf{X}\mathbf{w}^{(1)}\|_2} \gamma^{(1)} + \frac{\mathbf{1}}{\sqrt{n}}\alpha^{(1)} \right)_+ \right\|_2 \leq \beta \Longleftrightarrow \forall i \in [P], \ \exists \mathbf{S}_i \in \mathcal{K}_i \ \text{s.t.} \ \langle \mathbf{V}, \mathbf{D}_i \mathbf{U}' \mathbf{S}_i \rangle \leq \beta.$$

Similar arguments also hold for the negative side of the absolute value constraint. We then directly follow the steps in Sahiner et al. (2021a) and achieve the following convex program

$$\min_{\mathbf{S}_i} \frac{1}{2} \left\| \sum_{i=1}^{P} \mathbf{D}_i \mathbf{U}' \mathbf{S}_i - \mathbf{Y} \right\|_F^2 + \beta \sum_{i=1}^{P} \|\mathbf{S}_i\|_{c_i,*},$$

where

$$\|\mathbf{S}\|_{c_i,*} := \min_{t \geq 0} t \ \text{s.t.} \ \mathbf{S} \in t\mathcal{K}_i$$

denotes a constrained version of the standard nuclear norm. $\qquad\square$

## C.6 PROOF OF THEOREM 2.4

We first restate the dual problem as

$$\max_{\mathbf{V}} -\frac{1}{2}\|\mathbf{V} - \mathbf{Y}\|_F^2 + \frac{1}{2}\|\mathbf{Y}\|_F^2 \ \text{s.t.} \ \max_{\|\mathbf{s}'\|_2=1} \left\| \mathbf{V}^T \left( \mathbf{U}'\mathbf{s}' \right)_+ \right\|_2 \leq \beta, \tag{35}$$

where we use the results from Theorem 2.3 to modify the dual constraint. Since $\mathbf{Y}$ is one-hot encoded, (35) can be rewritten as

$$\max_{\mathbf{V}} -\frac{1}{2}\|\mathbf{V} - \mathbf{Y}\|_F^2 + \frac{1}{2}\|\mathbf{Y}\|_F^2 \ \text{s.t.} \ \max_{\|\mathbf{u}\|_2=1} \|\mathbf{V}^T\mathbf{u}\|_2 \leq \beta. \tag{36}$$

This problem has the following closed-form solution

$$\mathbf{v}_j^* = \begin{cases} \beta \frac{\mathbf{y}_j}{\|\mathbf{y}_j\|_2} & \text{if } \beta \leq \|\mathbf{y}_j\|_2 \\ \mathbf{y}_j & \text{otherwise} \end{cases}, \quad \forall j \in [C]. \tag{37}$$

and the corresponding first layer weights in (36) are

$$\mathbf{w}_j^{(1)^*} = \begin{cases} \mathbf{X}^\dagger \mathbf{y}_j & \text{if } \beta \leq \|\mathbf{y}_j\|_2 \\ - & \text{otherwise} \end{cases}, \quad \forall j \in [o]. \tag{38}$$

Now let us first denote the set of indices that achieves the extreme point of the dual constraint as $\mathcal{E} := \{j : \beta \leq \|\mathbf{y}_j\|_2, j \in [C]\}$. Then the dual objective in (36) using the optimal dual parameter in (37)

$$\begin{aligned} d_{2v}^* &= -\frac{1}{2}\|\mathbf{V}^* - \mathbf{Y}\|_F^2 + \frac{1}{2}\|\mathbf{Y}\|_F^2 \\ &= -\frac{1}{2}\sum_{j \in \mathcal{E}}(\beta - \|\mathbf{y}_j\|_2)^2 + \frac{1}{2}\sum_{j=1}^{C}\|\mathbf{y}_j\|_2^2 \\ &= -\frac{1}{2}\beta^2|\mathcal{E}| + \beta \sum_{j \in \mathcal{E}}\|\mathbf{y}_j\|_2 + \frac{1}{2}\sum_{j \notin \mathcal{E}}\|\mathbf{y}_j\|_2^2. \end{aligned} \tag{39}$$

We next restate the scaled primal problem

$$p_{2v}^* = \min_{\theta \in \Theta_s} \frac{1}{2} \left\| \sum_{j=1}^{m_1} \left( \frac{(\mathbf{I}_n - \frac{1}{n}\mathbf{1}\mathbf{1}^T)\mathbf{X}\mathbf{w}_j^{(1)}}{\|(\mathbf{I}_n - \frac{1}{n}\mathbf{1}\mathbf{1}^T)\mathbf{X}\mathbf{w}_j^{(1)}\|_2} \gamma_j^{(1)} + \frac{\mathbf{1}}{\sqrt{n}}\alpha_j^{(1)} \right)_+ \mathbf{w}_j^{(2)^T} - \mathbf{Y} \right\|_F^2 + \beta \sum_{j=1}^{m_1} \left\| \mathbf{w}_j^{(2)} \right\|_2 \tag{40}$$

and then obtain the optimal primal objective using the hidden layer weights in (38), which yields the following optimal solution

$$\left(\mathbf{w}_j^{(1)^*}, \mathbf{w}_j^{(2)^*}\right) = \begin{cases} \left(\mathbf{X}^\dagger \mathbf{y}_j, (\|\mathbf{y}_j\|_2 - \beta)\,\mathbf{e}_j\right) & \text{if } \beta \le \|\mathbf{y}_j\|_2 \\ - & \text{otherwise} \end{cases}$$

$$\begin{bmatrix} \gamma_j^{(1)^*} \\ \alpha_j^{(1)^*} \end{bmatrix} = \frac{1}{\|\mathbf{y}_j\|_2} \begin{bmatrix} \|\mathbf{y}_j - \frac{1}{n}\mathbf{1}\mathbf{1}^T\mathbf{y}_j\|_2 \\ \frac{1}{\sqrt{n}}\mathbf{1}^T\mathbf{y}_j \end{bmatrix} \tag{41}$$

We now evaluate the primal problem objective with the parameters in (41), which yields

$$
\begin{aligned}
p_{2v}^* &= \frac{1}{2} \left\| \sum_{j=1}^C \left( \frac{(\mathbf{I}_n - \frac{1}{n}\mathbf{1}\mathbf{1}^T)\mathbf{X}\mathbf{w}_j^{(1)^*}}{\|(\mathbf{I}_n - \frac{1}{n}\mathbf{1}\mathbf{1}^T)\mathbf{X}\mathbf{w}_j^{(1)^*}\|_2} \gamma_j^{(1)^*} + \frac{1}{\sqrt{n}}\alpha_j^{(1)^*} \right) \mathbf{w}_j^{(2)^{*T}} - \mathbf{Y} \right\|_{+}^2 \Bigg\|_F^2 + \beta \sum_{j=1}^C \left\| \mathbf{w}_j^{(2)^*} \right\|_2 \\
&= \frac{1}{2} \left\| \sum_{j \in \mathcal{E}} (\|\mathbf{y}_j\|_2 - \beta) \frac{\mathbf{y}_j}{\|\mathbf{y}_j\|_2}\mathbf{e}_j^T - \mathbf{Y} \right\|_F^2 + \beta \sum_{j \in \mathcal{E}} (\|\mathbf{y}_j\|_2 - \beta) \\
&= \frac{1}{2}\sum_{j \in \mathcal{E}} \left\| \beta \frac{\mathbf{y}_j}{\|\mathbf{y}_j\|_2}\mathbf{e}_j^T \right\|_F^2 + \frac{1}{2}\sum_{j \notin \mathcal{E}} \|\mathbf{y}_j\mathbf{e}_j^T\|_F^2 + \beta \sum_{j \in \mathcal{E}} \|\mathbf{y}_j\|_2 - \beta^2|\mathcal{E}| \\
&= -\frac{1}{2}\beta^2|\mathcal{E}| + \frac{1}{2}\sum_{j \notin \mathcal{E}} \|\mathbf{y}_j\|_2^2 + \beta \sum_{j \in \mathcal{E}} \|\mathbf{y}_j\|_2,
\end{aligned}
\tag{42}
$$

which is the same with (39). Therefore, strong duality holds, i.e., $p_{2v}^* = d_{2v}^*$, and the layer weights in (41) are optimal for the primal problem (40). $\qquad\square$

# D    CONVOLUTIONAL NEURAL NETWORKS

## D.1    PROOF OF THEOREM 3.1

Following the arguments in Theorem 2.2 and Ergen & Pilanci (2021a), strong duality holds for the CNN training problem in (13), i.e., $p_{2c}^* = d_{2c}^*$, and the dual constraint can be equivalently written as follows

$$
\begin{aligned}
&\max_{\substack{\mathbf{w}^{(1)} \\ \gamma^{(1)2}+\alpha^{(1)2}=1}} \left| \sum_{k=1}^K \mathbf{v}^T \left( \frac{\mathbf{X}_k\mathbf{z} - \mu\mathbf{1}}{\sqrt{\sum_{k'=1}^K \|\mathbf{X}_{k'}\mathbf{z} - \mu\mathbf{1}\|_2^2}}\gamma^{(1)} + \frac{1}{\sqrt{nK}}\mathbf{1}\alpha^{(1)} \right)_+ \right| \\
&= \max_{\theta \in \Theta_s} \left| \mathbf{v}^T\mathbf{C} \left( \frac{(\mathbf{I}_{nK} - \frac{1}{nK}\mathbf{1}\mathbf{1}^T)\mathbf{M}\mathbf{z}}{\|(\mathbf{I}_{nK} - \frac{1}{nK}\mathbf{1}\mathbf{1}^T)\mathbf{M}\mathbf{z}\|_2}\gamma^{(1)} + \frac{1}{\sqrt{nK}}\alpha^{(1)} \right)_+ \right| \\
&= \max_{\|\mathbf{q}\|_2=\|\mathbf{s}\|_2=1} \left| \mathbf{v}^T\mathbf{C} \left( \begin{bmatrix} \mathbf{U}_M\mathbf{q} & \frac{1}{\sqrt{nK}} \end{bmatrix}\mathbf{s} \right)_+ \right| \\
&= \max_{\|\mathbf{s}'\|_2=1} \left| \mathbf{v}^T\mathbf{C} \left( \mathbf{U}_M'\mathbf{s}' \right)_+ \right| \\
&= \max_{\|\mathbf{s}'\|_2=1} \left| \sum_{k=1}^K \mathbf{v}^T \left( \mathbf{U}_{Mk}'\mathbf{s}' \right)_+ \right| \\
&= \max_{i \in [P]} \max_{\substack{\|\mathbf{s}'\|_2=1 \\ (2\mathbf{D}_{ik}-\mathbf{I}_n)\mathbf{U}_{Mk}'\mathbf{s}' \ge 0}} \left| \sum_{k=1}^K \mathbf{v}^T\mathbf{D}_{ik}\mathbf{U}_{Mk}'\mathbf{s}' \right|,
\end{aligned}
$$

where $\mathbf{C} := \mathbf{1}_K^T \otimes \mathbf{I}_n$, $\mathbf{M} = [\mathbf{X}_1; \mathbf{X}_2; \dots \mathbf{X}_K]$, and we use the following variable changes $\mathbf{q} := \boldsymbol{\Sigma}_M\mathbf{V}_M^T\mathbf{z}$, $\mathbf{s} := \begin{bmatrix} \gamma^{(1)} & \alpha^{(1)} \end{bmatrix}^T$, $\mathbf{s}' := \begin{bmatrix} \gamma^{(1)}\mathbf{q}^T & \alpha^{(1)} \end{bmatrix}^T$, and $\mathbf{U}_M' := \begin{bmatrix} \mathbf{U}_M & \frac{1}{\sqrt{nK}} \end{bmatrix} = [\mathbf{U}_{M1}'; \mathbf{U}_{M2}'; \dots \mathbf{U}_{MK}']$.

Then, following the steps in Theorem 2.2 and Ergen & Pilanci (2021a), the equivalent convex program for (13) is

$$\min_{\mathbf{q}_i, \mathbf{q}'_i} \frac{1}{2} \left\| \sum_{i=1}^{P_c} \sum_{k=1}^{K} \mathbf{D}_{ik} \mathbf{U}'_{Mk} (\mathbf{q}_i - \mathbf{q}'_i) \right\| + \beta \sum_{i=1}^{P_c} (\|\mathbf{q}_i\|_2 + \|\mathbf{q}'_i\|_2)$$
$$\text{s.t. } (2\mathbf{D}_{ik} - \mathbf{I}_n) \mathbf{U}'_{Mk} \mathbf{q}_i \geq 0, \ (2\mathbf{D}_{ik} - \mathbf{I}_n) \mathbf{U}'_{Mk} \mathbf{q}'_i \geq 0, \ \forall i \in [P_c], \forall k \in [K].$$

$\square$

# E  THE IMPLICIT REGULARIZATION EFFECT OF SGD ON BN NETWORKS

## E.1  PROOF OF LEMMA 5.1

We start with the following non-convex training problem

$$p^*_{v2} := \min_{\theta \in \Theta} \frac{1}{2} \left\| \sum_{j=1}^{m_1} \left( \text{BN}_{\gamma, \alpha} \left( \mathbf{X} \mathbf{w}_j^{(1)} \right) \right)_+ \mathbf{w}_j^{(2)^T} - \mathbf{Y} \right\|_F^2 + \frac{\beta}{2} \sum_{j=1}^{m_1} \left( \gamma_j^{(1)^2} + \alpha_j^{(1)^2} + \left\| \mathbf{w}_j^{(2)} \right\|_2^2 \right),$$

where we note that hidden layer weights are not regularized without loss of generality due to the arguments in Appendix B.2.1. Now, consider the SVD of the zero mean form of the data matrix $(\mathbf{I}_n - \frac{1}{n} \mathbf{1} \mathbf{1}^\top) \mathbf{X} = \mathbf{U} \boldsymbol{\Sigma} \mathbf{V}^\top$. We can thus equivalently write the problem as

$$p^*_{v2} = \min_{\theta \in \Theta} \frac{1}{2} \left\| \sum_{j=1}^{m_1} \left( \frac{\mathbf{U} \boldsymbol{\Sigma} \mathbf{V}^\top \mathbf{w}_j^{(1)}}{\|\mathbf{U} \boldsymbol{\Sigma} \mathbf{V}^\top \mathbf{w}_j^{(1)}\|_2} \gamma_j^{(1)} + \frac{\mathbf{1}}{\sqrt{n}} \alpha_j^{(1)} \right)_+ \mathbf{w}_j^{(2)^T} - \mathbf{Y} \right\|_F^2 + \frac{\beta}{2} \sum_{j=1}^{m_1} \left( \gamma_j^{(1)^2} + \alpha_j^{(1)^2} + \left\| \mathbf{w}_j^{(2)} \right\|_2^2 \right),$$

Now, simply let $\mathbf{q}_j = \boldsymbol{\Sigma} \mathbf{V}^\top \mathbf{w}_j^{(1)}$. Noting that $\|\mathbf{U} \mathbf{q}_j\|_2 = \|\mathbf{q}_j\|_2$, we arrive at

$$p^*_{v2} = p^*_{v2b} := \min_{\theta \in \Theta} \frac{1}{2} \left\| \sum_{j=1}^{m_1} \left( \frac{\mathbf{U} \mathbf{q}_j}{\|\mathbf{q}_j\|_2} \gamma_j^{(1)} + \frac{\mathbf{1}}{\sqrt{n}} \alpha_j^{(1)} \right)_+ \mathbf{w}_j^{(2)^T} - \mathbf{Y} \right\|_F^2 + \frac{\beta}{2} \sum_{j=1}^{m_1} \left( \gamma_j^{(1)^2} + \alpha_j^{(1)^2} + \left\| \mathbf{w}_j^{(2)} \right\|_2^2 \right),$$

$\square$

## E.2  PROOF OF THEOREM 5.1

When solving (9) with GD, we have the updates

$$\mathbf{w}_j^{(1)^{(k+1)}} = \mathbf{w}_j^{(1)^{(k)}} - \eta_k \nabla_{\mathbf{w}_j^{(1)}} \mathcal{L}(f_{v2}^k(\mathbf{X}), \mathbf{Y})$$

$$\alpha_j^{(1)^{(k+1)}} = \alpha_j^{(1)^{(k)}} - \eta_k \left( \nabla_{\alpha_j^{(1)}} \mathcal{L}(f_{v2}^k(\mathbf{X}), \mathbf{Y}) + \beta \alpha_j^{(1)} \right)$$

$$\gamma_j^{(1)^{(k+1)}} = \gamma_j^{(1)^{(k)}} - \eta_k \left( \nabla_{\gamma_j^{(1)}} \mathcal{L}(f_{v2}^k(\mathbf{X}), \mathbf{Y}) + \beta \gamma_j^{(1)} \right)$$

$$\mathbf{w}_j^{(2)^{(k+1)}} = \mathbf{w}_j^{(2)^{(k)}} - \eta_k \left( \nabla_{\mathbf{w}_j^{(2)}} \mathcal{L}(f_{v2}^k(\mathbf{X}), \mathbf{Y}) + \beta \mathbf{w}_j^{(2)} \right)$$

Now, let $\mathbf{D}_j^{(k)} = \text{diag}\{\mathbb{1}[\mathbf{X}\mathbf{w}_j^{(1)\,(k)} \geq 0]\}$, $\mathbf{R}^{(k)} = \sum_{j=1}^{m_1}\left(\text{BN}_{\gamma,\alpha}\left(\mathbf{X}\mathbf{w}_j^{(1)\,(k)}\right)\right)_+ \mathbf{w}_j^{(2)\,(k)^T} - \mathbf{Y}$ be

the residual at step $k$, and $\mathbf{X}^0 = (\mathbf{I}_n - \frac{1}{n}\mathbf{1}\mathbf{1}^\top)\mathbf{X}$. Then, with squared loss $\mathcal{L}$, we have the following:

$$\nabla_{\mathbf{w}_j^{(1)}}\mathcal{L}(f_{v2}^k(\mathbf{X}),\mathbf{Y}) = \frac{\gamma_j^{(1)\,(k)}}{\|\mathbf{X}^0\mathbf{w}_j^{(1)\,(k)}\|_2}(\mathbf{D}_j^{(k)}\mathbf{X}^0)^\top\mathbf{R}^{(k)}\mathbf{w}_j^{(2)\,(k)}$$

$$- \frac{\gamma_j^{(1)\,(k)}}{\|\mathbf{X}^0\mathbf{w}_j^{(1)\,(k)}\|_2^3}\left(\mathbf{w}_j^{(1)\,(k)^\top}(\mathbf{D}_j^{(k)}\mathbf{X}^0)^\top\mathbf{R}^{(k)}\mathbf{w}_j^{(2)\,(k)}\right)\mathbf{X}^{0\top}\mathbf{X}^0\mathbf{w}_j^{(1)\,(k)}$$

$$\nabla_{\alpha_j^{(1)}}\mathcal{L}(f_{v2}^k(\mathbf{X}),\mathbf{Y}) = \mathbf{w}_j^{(2)\,(k)^T}\mathbf{R}^{(k)^\top}\mathbf{D}_j^{(k)}\mathbf{1}$$

$$\nabla_{\gamma_j^{(1)}}\mathcal{L}(f_{v2}^k(\mathbf{X}),\mathbf{Y}) = \frac{1}{\|\mathbf{X}^0\mathbf{w}_j^{(1)\,(k)}\|_2}\mathbf{w}_j^{(2)\,(k)^T}\mathbf{R}^{(k)^\top}\mathbf{D}_j^{(k)}\mathbf{X}^0\mathbf{w}_j^{(1)\,(k)}$$

$$\nabla_{\mathbf{w}_j^{(2)}}\mathcal{L}(f_{v2}^k(\mathbf{X}),\mathbf{Y}) = \mathbf{R}^{(k)^\top}\mathbf{D}_j^{(k)}\left(\frac{\mathbf{X}^0\mathbf{w}_j^{(1)\,(k)}}{\|\mathbf{X}^0\mathbf{w}_j^{(1)\,(k)}\|_2}\gamma_j^{(1)\,(k)} + \frac{1}{\sqrt{n}}\alpha_j^{(1)}\right)$$

To express the updates in $\mathbf{q}_j$-space for $\mathbf{w}_j^{(1)}$, we can compute

$$\Delta_{v2,j}^{(k)} = \boldsymbol{\Sigma}\mathbf{V}^\top\nabla_{\mathbf{w}_j^{(1)}}\mathcal{L}(f_{v2}^k(\mathbf{X}),\mathbf{Y})$$

$$= \frac{\gamma_j^{(1)\,(k)}}{\|\mathbf{X}^0\mathbf{w}_j^{(1)\,(k)}\|_2}\boldsymbol{\Sigma}^2\mathbf{U}^\top\mathbf{D}_j^{(k)}\mathbf{R}^{(k)}\mathbf{w}_j^{(2)\,(k)} - \frac{\gamma_j^{(1)\,(k)}}{\|\mathbf{X}^0\mathbf{w}_j^{(1)\,(k)}\|_2^3}\left(\mathbf{w}_j^{(1)\,(k)^\top}\mathbf{X}^{0\top}\mathbf{D}_j^{(k)}\mathbf{R}^{(k)}\mathbf{w}_j^{(2)\,(k)}\right)\boldsymbol{\Sigma}^2\mathbf{U}^\top\mathbf{X}^0\mathbf{w}_j^{(1)\,(k)}$$

Now, we consider solving (16) with GD. We assume that all weights are equivalent to those in (9), i.e.
$f_{v2b}^k(\mathbf{X}) = f_{v2}^k(\mathbf{X})$, and $\mathbf{q}_j^{(k)} = \boldsymbol{\Sigma}\mathbf{V}^\top\mathbf{w}_j^{(1)\,(k)}$. In this case, the gradients are given by:

$$\nabla_{\mathbf{q}_j}\mathcal{L}(f_{v2b}^k(\mathbf{X}),\mathbf{Y}) = \frac{\gamma_j^{(1)\,(k)}}{\|\mathbf{q}_j^{(k)}\|_2}(\mathbf{D}_j^{(k)}\mathbf{U})^\top\mathbf{R}^{(k)}\mathbf{w}_j^{(2)\,(k)} - \frac{\gamma_j^{(1)\,(k)}}{\|\mathbf{q}_j^{(k)}\|_2^3}\left(\mathbf{q}_j^{(k)^\top}(\mathbf{D}_j^{(k)}\mathbf{U})^\top\mathbf{R}^{(k)}\mathbf{w}_j^{(2)\,(k)}\right)\mathbf{q}_j^{(k)}$$

$$\nabla_{\alpha_j^{(1)}}\mathcal{L}(f_{v2b}^k(\mathbf{X}),\mathbf{Y}) = \mathbf{w}_j^{(2)\,(k)^T}\mathbf{R}^{(k)^\top}\mathbf{D}_j^{(k)}\mathbf{1}$$

$$\nabla_{\gamma_j^{(1)}}\mathcal{L}(f_{v2b}^k(\mathbf{X}),\mathbf{Y}) = \frac{1}{\|\mathbf{q}_j^{(k)}\|_2}\mathbf{w}_j^{(2)\,(k)^T}\mathbf{R}^{(k)^\top}\mathbf{D}_j^{(k)}\mathbf{U}\mathbf{q}_j^{(k)}$$

$$\nabla_{\mathbf{w}_j^{(2)}}\mathcal{L}(f_{v2b}^k(\mathbf{X}),\mathbf{Y}) = \mathbf{R}^{(k)^\top}\mathbf{D}_j^{(k)}\left(\frac{\mathbf{U}\mathbf{q}_j^{(k)}}{\|\mathbf{q}_j^{(k)}\|_2}\gamma_j^{(1)\,(k)} + \frac{1}{\sqrt{n}}\alpha_j^{(1)}\right)$$

Simplifying, we have

$$\Delta_{v2b,j}^{(k)} = \frac{\gamma_j^{(1)\,(k)}}{\|\mathbf{q}_j^{(k)}\|_2}\mathbf{U}^\top\mathbf{D}_j^{(k)}\mathbf{R}^{(k)}\mathbf{w}_j^{(2)\,(k)} - \frac{\gamma_j^{(1)\,(k)}}{\|\mathbf{q}_j^{(k)}\|_2^3}\left(\mathbf{q}_j^{(k)^\top}\mathbf{U}^\top\mathbf{D}_j^{(k)}\mathbf{R}^{(k)}\mathbf{w}_j^{(2)\,(k)}\right)\mathbf{q}_j^{(k)}$$

Now, noting that $\mathbf{X}^0\mathbf{w}_j^{(1)\,(k)} = \mathbf{U}\mathbf{q}_j^{(k)}$, and thus that $\|\mathbf{X}^0\mathbf{w}_j^{(1)\,(k)}\|_2 = \|\mathbf{q}_j^{(k)}\|_2$, we can find that, as desired,

$$\Delta_{v2b,j}^{(k)} = \boldsymbol{\Sigma}^2\Delta_{v2,j}^{(k)}$$

$$\nabla_{\gamma_j^{(1)}}\mathcal{L}(f_{v2b}^k(\mathbf{X}),\mathbf{Y}) = \nabla_{\gamma_j^{(1)}}\mathcal{L}(f_{v2}^k(\mathbf{X}),\mathbf{Y})$$

$$\nabla_{\alpha_j^{(1)}}\mathcal{L}(f_{v2b}^k(\mathbf{X}),\mathbf{Y}) = \nabla_{\alpha_j^{(1)}}\mathcal{L}(f_{v2}^k(\mathbf{X}),\mathbf{Y})$$

$$\nabla_{\mathbf{w}_j^{(2)}}\mathcal{L}(f_{v2b}^k(\mathbf{X}),\mathbf{Y}) = \nabla_{\mathbf{w}_j^{(2)}}\mathcal{L}(f_{v2}^k(\mathbf{X}),\mathbf{Y})$$

$\square$

# F  DEEP NETWORKS

## F.1  PROOF OF THEOREM 4.1

Applying the same steps in Theorem 2.1 for the data matrix $\mathbf{A}^{(L-2)}$, i.e., replacing $\mathbf{X}$ with $\mathbf{A}^{(L-2)}$, yields the following optimal parameters

$$
\mathbf{W}^{(1)^*} = \begin{cases}
\left(\mathbf{A}^{(L-2)^\dagger}(\mathbf{y})_+, \mathbf{A}^{(L-2)^\dagger}(-\mathbf{y})_+\right) & \text{if } \beta \leq \|(\mathbf{y})_+\|_2, \beta \leq \|(-\mathbf{y})_+\|_2 \\
\mathbf{A}^{(L-2)^\dagger}(-\mathbf{y})_+ & \text{if } \beta > \|(\mathbf{y})_+\|_2, \beta \leq (-\mathbf{y})_+\|_2 \\
\mathbf{A}^{(L-2)^\dagger}(\mathbf{y})_+ & \text{if } \beta \leq \|(\mathbf{y})_+\|_2, \beta > \|(\mathbf{y})_+\|_2 \\
\mathbf{0} & \text{if } \beta > \|(\mathbf{y})_+\|_2, \beta > \|(-\mathbf{y})_+\|_2
\end{cases}
$$

$$
\begin{bmatrix} \gamma_1^{(1)^*} \\ \alpha_1^{(1)^*} \end{bmatrix} = \frac{1}{\|(\mathbf{y})_+\|_2} \begin{bmatrix} \|(\mathbf{y})_+ - \frac{1}{n}\mathbf{1}\mathbf{1}^T(\mathbf{y})_+\|_2 \\ \frac{1}{\sqrt{n}}\mathbf{1}^T(\mathbf{y})_+ \end{bmatrix} \qquad
\begin{bmatrix} \gamma_2^{(1)^*} \\ \alpha_2^{(1)^*} \end{bmatrix} = \frac{1}{\|(-\mathbf{y})_+\|_2} \begin{bmatrix} \|(-\mathbf{y})_+ - \frac{1}{n}\mathbf{1}\mathbf{1}^T(-\mathbf{y})_+\|_2 \\ \frac{1}{\sqrt{n}}\mathbf{1}^T(-\mathbf{y})_+ \end{bmatrix}
$$

$$
\mathbf{w}^{(2)^*} = \begin{cases}
\begin{bmatrix} \|(\mathbf{y})_+\|_2 - \beta \\ -\|(-\mathbf{y})_+\|_2 + \beta \end{bmatrix} & \text{if } \beta \leq \|(\mathbf{y})_+\|_2, \beta \leq \|(-\mathbf{y})_+\|_2 \\
-\|(-\mathbf{y})_+\|_2 + \beta & \text{if } \beta > \|(\mathbf{y})_+\|_2, \beta \leq \|(-\mathbf{y})_+\|_2 \\
\|(\mathbf{y})_+\|_2 - \beta & \text{if } \beta \leq \|(\mathbf{y})_+\|_2, \beta > \|(-\mathbf{y})_+\|_2 \\
0 & \text{if } \beta > \|(\mathbf{y})_+\|_2, \beta > \|(-\mathbf{y})_+\|_2
\end{cases}
$$

$$
\mathbf{v}^* = \begin{cases}
\beta \frac{(\mathbf{y})_+}{\|(\mathbf{y})_+\|_2} - \beta \frac{(-\mathbf{y})_+}{\|(-\mathbf{y})_+\|_2} & \text{if } \beta \leq \|(\mathbf{y})_+\|_2, \beta \leq \|(-\mathbf{y})_+\|_2 \\
(\mathbf{y})_+ - \beta \frac{(-\mathbf{y})_+}{\|(-\mathbf{y})_+\|_2} & \text{if } \beta > \|(\mathbf{y})_+\|_2, \beta \leq (-\mathbf{y})_+\|_2 \\
\beta \frac{(\mathbf{y})_+}{\|(\mathbf{y})_+\|_2} - (-\mathbf{y})_+ & \text{if } \beta \leq \|(\mathbf{y})_+\|_2, \beta > \|(-\mathbf{y})_+\|_2 \\
\mathbf{y} & \text{if } \beta > \|(\mathbf{y})_+\|_2, \beta > \|(-\mathbf{y})_+\|_2
\end{cases}.
$$

We now substitute these parameters into the primal (3) and the dual (5) to prove that strong duality holds. For presentation simplicity, we only consider the first case, i.e., $\beta \leq \|(\mathbf{y})_+\|_2, \beta \leq \|(-\mathbf{y})_+\|_2$ and note that the same derivations apply to the other cases as well. We first evaluate the primal objective with these parameters as follows

$$
\begin{aligned}
p_L^* &= \frac{1}{2} \left\| \sum_{j=1}^2 \left( \frac{(\mathbf{I}_n - \frac{1}{n}\mathbf{1}\mathbf{1}^T)\mathbf{A}^{(L-2)}\mathbf{w}_j^{(L-1)^*}}{\|(\mathbf{I}_s - \frac{1}{n}\mathbf{1}\mathbf{1}^T)\mathbf{A}^{(L-2)}\mathbf{w}_j^{(L-1)^*}\|_2} \gamma_j^{(l)} + \frac{1}{\sqrt{n}}\alpha_j^{(l)^*} \right)_+ w_j^{(L)^*} - \mathbf{y} \right\|_2^2 + \beta \left\| \mathbf{w}^{(L)^*} \right\|_1 \\
&= \frac{1}{2} \left\| -\beta \frac{(\mathbf{y})_+}{\|(\mathbf{y})_+\|_2} + \beta \frac{(-\mathbf{y})_+}{\|(-\mathbf{y})_+\|_2} \right\|_2^2 + \beta(\|(\mathbf{y})_+\|_2 + \|(-\mathbf{y})_+\|_2 - 2\beta) \\
&= \beta(\|(\mathbf{y})_+\|_2 + \|(-\mathbf{y})_+\|_2) - \beta^2.
\end{aligned}
\tag{43}
$$

We then evaluate the dual objective as follows

$$
\begin{aligned}
d_L^* &= -\frac{1}{2}\|\mathbf{v}^* - \mathbf{y}\|_2^2 + \frac{1}{2}\|\mathbf{y}\|_2^2 \\
&= -\frac{1}{2} \left\| \beta \frac{(\mathbf{y})_+}{\|(\mathbf{y})_+\|_2} - \beta \frac{(-\mathbf{y})_+}{\|(-\mathbf{y})_+\|_2} - \mathbf{y} \right\|_2^2 + \frac{1}{2}\|\mathbf{y}\|_2^2 \\
&= -\beta^2 + \beta(\|(\mathbf{y})_+\|_2 + \|(-\mathbf{y})_+\|_2).
\end{aligned}
\tag{44}
$$

By (43) and (44), the set of parameters $\{\mathbf{W}^{(L-1)^*}, \gamma^{(L-1)^*}, \alpha^{(L-1)^*}, \mathbf{w}^{(L)^*}\}$ achieves $p_L^* = d_L^*$, therefore, strong duality holds. □

### F.2 PROOF OF THEOREM 4.2

Applying the same steps in Theorem 2.4 for the data matrix $\mathbf{A}^{(L-2)}$, i.e., replacing $\mathbf{X}$ with $\mathbf{A}^{(L-2)}$, yields the following set of parameters

$$\left(\mathbf{w}_j^{(L-1)^*}, \mathbf{w}_j^{(L)^*}\right) = \begin{cases} \left(\mathbf{A}^{(L-2)^\dagger}\mathbf{y}_j, (\|\mathbf{y}_j\|_2 - \beta)\,\mathbf{e}_j\right) & \text{if } \beta \leq \|\mathbf{y}_j\|_2 \\ \qquad\qquad - & \text{otherwise} \end{cases}$$

$$\begin{bmatrix} \gamma_j^{(L-1)^*} \\ \alpha_j^{(L-1)^*} \end{bmatrix} = \frac{1}{\|\mathbf{y}_j\|_2} \begin{bmatrix} \|\mathbf{y}_j - \frac{1}{n}\mathbf{1}\mathbf{1}^T\mathbf{y}_j\|_2 \\ \frac{1}{\sqrt{n}}\mathbf{1}^T\mathbf{y}_j \end{bmatrix} \qquad\qquad , \quad \forall j \in [C].$$

$$\mathbf{v}_j^* = \begin{cases} \beta\frac{\mathbf{y}_j}{\|\mathbf{y}_j\|_2} & \text{if } \beta \leq \|\mathbf{y}_j\|_2 \\ \mathbf{y}_j & \text{otherwise} \end{cases}$$

$\square$

## G  BN AFTER RELU

Despite the common practice for BN, i.e., placing BN layers before activation functions, some recent studies show that applying BN after activation function might be more effective (Chen et al., 2019). Thus, we analyze the effects of BN when it is employed after ReLU activations.

After applying the scaling in Lemma 1.1, the primal training problem is as follows

$$\min_{\theta \in \Theta_s} \frac{1}{2} \left\| \sum_{j=1}^{m_1} \mathrm{BN}_{\gamma,\alpha}\left(\left(\mathbf{X}\mathbf{w}_j^{(1)}\right)_+\right) w_j^{(2)} - \mathbf{y} \right\|_2^2 + \beta\|\mathbf{w}^{(2)}\|_1. \tag{45}$$

In the sequel, we repeat the same analysis in Section 2.

**Theorem G.1.** *For any data matrix $\mathbf{X}$, the non-convex training problem in (45) can be equivalently stated as the following finite-dimensional convex program*

$$\min_{\mathbf{s}_i,\mathbf{s}_i'} \frac{1}{2} \left\| \sum_{i=1}^P \mathbf{U}_i'(\mathbf{s}_i - \mathbf{s}_i') - \mathbf{y} \right\|_2^2 + \beta \sum_{i=1}^P (\|\mathbf{s}_i\|_2 + \|\mathbf{s}_i'\|_2) \; s.t. \; \begin{array}{l} (2\mathbf{D}_i - \mathbf{I}_n)\mathbf{X}\mathbf{V}_i\mathbf{\Sigma}_i^{-1}\mathbf{s}_{i,1:r_i-1} \geq 0 \\ (2\mathbf{D}_i - \mathbf{I}_n)\mathbf{X}\mathbf{V}_i\mathbf{\Sigma}_i^{-1}\mathbf{s}_{i,1:r_i-1}' \geq 0 \end{array}, \forall i \tag{46}$$

*provided that $(\mathbf{I}_n - \frac{1}{n}\mathbf{1}\mathbf{1}^T)\mathbf{D}_i\mathbf{X} := \mathbf{U}_i\mathbf{\Sigma}_i\mathbf{V}_i^T$, $\mathbf{U}_i' := \begin{bmatrix} \mathbf{U}_i & \frac{1}{\sqrt{n}}\mathbf{1} \end{bmatrix}$, $r_i := \mathrm{rank}((\mathbf{I}_n - \frac{1}{n}\mathbf{1}\mathbf{1}^T)\mathbf{D}_i\mathbf{X})$, and $\mathbf{s}_{i,1:r_i-1}$ denotes all the entries of $\mathbf{s}_i$ except the last one.*

**Theorem G.2.** *Suppose $n \leq d$ and $\mathbf{X}$ is full row-rank, then an optimal solution to (45) is given as*

$$\left(\mathbf{w}^{(1)^*}, w^{(2)^*}\right) = \left(\mathbf{X}^\dagger(\mathbf{y} - \min_i y_i), (\|\mathbf{y}\|_2 - \beta)_+\right), \begin{bmatrix} \gamma^{(1)^*} \\ \alpha^{(1)^*} \end{bmatrix} = \frac{1}{\|\mathbf{y}\|_2} \begin{bmatrix} \|\mathbf{y} - \frac{1}{n}\mathbf{1}\mathbf{1}^T\mathbf{y}\|_2 \\ \frac{1}{\sqrt{n}}\mathbf{1}^T\mathbf{y} \end{bmatrix}.$$

These results indicate that applying BN after ReLU leads to a completely different phenomena. Specifically, Theorem G.1 shows that the hyperplane arrangement matrices $\mathbf{D}_i\mathbf{X}$ are de-meaned and whitened via SVD, namely $(\mathbf{I}_n - \frac{1}{n}\mathbf{1}\mathbf{1}^T)\mathbf{D}_i\mathbf{X} = \mathbf{U}_i\mathbf{\Sigma}_i\mathbf{V}_i^T$. This renders whitened features $\mathbf{U}_i' = \begin{bmatrix} \mathbf{U}_i & 1/\sqrt{n} \end{bmatrix}$ appearing in the convex model. This is in contrast with applying BN before the ReLU layer, which directly operates on the data matrix $\mathbf{X}$.

Theorem G.2 shows that an optimal solution can be found in closed-form in the high-dimensional regime $n \leq d$ when $\mathbf{X}$ is full row-rank, which is typically satisfied in sample-limited scenarios. Surprisingly, a single neuron optimizes the training cost, which corresponds to a single non-zero block in the group $\ell_1$ penalized program (46).

### G.1  PROOF OF THEOREM G.2

Applying the approach in Lemma C.1 to the following constraint

$$\max_{\theta \in \Theta_s} \left| \mathbf{v}^T \frac{(\mathbf{I}_n - \frac{1}{n}\mathbf{1}\mathbf{1}^T)\left(\mathbf{X}\mathbf{w}^{(1)}\right)_+}{\|(\mathbf{I}_n - \frac{1}{n}\mathbf{1}\mathbf{1}^T)\left(\mathbf{X}\mathbf{w}^{(1)}\right)_+\|_2}\gamma^{(1)} + \alpha^{(1)}\mathbf{v}^T\frac{\mathbf{1}}{\sqrt{n}} \right|$$

yields

$$\mathbf{v}^* = \beta \frac{\mathbf{y}}{\|\mathbf{y}\|_2}$$

Then, the rest of proof is to verify the solutions by substituting them into the primal and dual problems as in Theorem 2.1. $\square$

### G.2 PROOF OF THEOREM G.1

We first focus on the dual constraint as follows

$$\max_{\substack{\mathbf{w}^{(1)} \\ \gamma^{(1)2}+\alpha^{(1)2}=1}} \left| \frac{\mathbf{v}^T(\mathbf{I}_n - \frac{1}{n}\mathbf{1}\mathbf{1}^T)\left(\mathbf{X}\mathbf{w}^{(1)}\right)_+}{\|(\mathbf{I}_n - \frac{1}{n}\mathbf{1}\mathbf{1}^T)\left(\mathbf{X}\mathbf{w}^{(1)}\right)_+\|_2}\gamma^{(1)} + \alpha^{(1)}\frac{\mathbf{v}^T\mathbf{1}}{\sqrt{n}} \right|$$

$$= \max_{i\in[P]} \max_{\substack{\theta\in\Theta_s \\ (2\mathbf{D}_i-\mathbf{I}_n)\mathbf{X}\mathbf{w}^{(1)}\geq 0}} \left| \frac{\mathbf{v}^T(\mathbf{I}_n - \frac{1}{n}\mathbf{1}\mathbf{1}^T)\mathbf{D}_i\mathbf{X}\mathbf{w}^{(1)}}{\|(\mathbf{I}_n - \frac{1}{n}\mathbf{1}\mathbf{1}^T)\mathbf{D}_i\mathbf{X}\mathbf{w}^{(1)}\|_2}\gamma^{(1)} + \alpha^{(1)}\frac{\mathbf{v}^T\mathbf{1}}{\sqrt{n}} \right|$$

$$= \max_{i\in[P]} \max_{\substack{\mathbf{q}_i,\alpha,\gamma \\ \alpha^2+\gamma^2=1 \\ (2\mathbf{D}_i-\mathbf{I}_n)\mathbf{X}\mathbf{V}_i\boldsymbol{\Sigma}_i^{-1}\mathbf{q}_i\geq 0}} \left| \mathbf{v}^T\frac{\mathbf{U}_i\mathbf{q}_i}{\|\mathbf{q}_i\|_2}\gamma + \alpha\mathbf{v}^T\frac{\mathbf{1}}{\sqrt{n}} \right|$$

$$= \max_{i\in[P]} \max_{\substack{\mathbf{s}_i \\ \|\mathbf{s}_i\|_2=1 \\ (2\mathbf{D}_i-\mathbf{I}_n)\mathbf{X}\mathbf{V}_i\boldsymbol{\Sigma}_i^{-1}\mathbf{s}_{1:r_i-1}\geq 0}} \left| \mathbf{v}^T\underbrace{\left[\mathbf{U}_i \quad \frac{\mathbf{1}}{\sqrt{n}}\right]}_{\mathbf{U}_i'}\mathbf{s} \right|.$$

Then, the rest of the proof directly follow from Theorem 2.2. $\square$

