# OpenReview forum: "Demystifying Batch Normalization in ReLU Networks: Equivalent Convex Optimization Models and Implicit Regularization"
_ICLR.cc/2022/Conference — ICLR 2022 Poster_

### Official Review · Reviewer_YR4A · 2021-11-02

**Correctness:** 3
**Technical Novelty And Significance:** 3
**Empirical Novelty And Significance:** 3
**Recommendation:** 6
**Confidence:** 3

**Main Review:**

The contribution of the paper is to show that regularized ReLU networks with batch normalization can be equivalently stated as convex problem. While there has been some related work investigating the benefits of batch normalization on neural network training, the present work provides a theoretical characterization for training deep networks with batch normalization.

Moreover, the paper provides an alternative algorithm via a convex program which is shown to perform well compared to the standard non-convex optimization. Since batch normalization is widely used in practice, this result may be of great interest to the community. Therefore I vote for acceptance.

As a minor remark, I found the structure as well as the headings of Section 4 a bit misleading, since first one gets the impression that its purpose is to point out a property of SGD applied to the BN network, while in fact its goal is to fix a weakness of the proposed approach, namely that it tends to generalize poorly. Maybe this could be made more explicit, in particular also reflected in the heading of the section as well as the discussion in the introduction. Furthermore, maybe the ordering of the sections can be improved. Currently Section 4 is placed between the two sections about the extensions to CNNs and the extension to the L-layer case. The flow of the paper could be improved by first covering the extensions to different architectures and then discuss the regularization issues (or the other way round). Moreover, throughout the paper, the abbreviations SGD and GD are used interchangeably.

**Summary Of The Paper:**

The paper studies batch normalization in deep neural networks. For a two-layer network with scalar output and batch normalization, a dual of the problem is derived. It is then shown that in the high-dimensional regime, the dual can be further simplified so that an optimal solution can be computed in closed form. In the general case, the concept of hyperplane arrangements can be used to formulate an equivalent finite-dimensional convex program which can be solved in polynomial time.

The analysis is then extended for vector output networks and different architectures including L-layer neural networks and CNNs. It is then shown that the convex problems tend to fit low singular value directions which will lead to poorer generalization. As a remedy, the authors propose a truncated variant of the problem by obtaining a low-rank approximation of the data.

In the experimental section, it is shown on the CIFAR-100 dataset that the closed form solution leads to superior results compared to the solution obtained by gradient descent. Moreover it is shown that the truncated variant of the approach is needed to achieve a good generalization performance.

**Summary Of The Review:**

Overall, despite some flaws in presentation, I think the paper poses a meaningful contribution which warrants acceptance.

---

### Official Review · Reviewer_pM4k · 2021-11-03

**Correctness:** 4
**Technical Novelty And Significance:** 3
**Empirical Novelty And Significance:** 3
**Recommendation:** 6
**Confidence:** 4

**Main Review:**

Overall I think the results presented in this work are solid and interesting.

**Pros**:

- The paper is well-written and the theory developed is solid and well-organized.

- The author(s) provided a quite comprehensive study (scalar output, vector output, feed-forward NN, convolutional NN, implicit regularization of SGD with BN) of batch normalization from the lens of convex duality.

- Experiments are also quite complete and well-designed.

**Cons**:
- Missing related work: [1, 2], and maybe some other subsequent papers that cited these works.
- In Theorem 2.2. $P$ grows exponentially, the convexity does not come free. Transforming a nonconvex problem into a convex problem by introducing more variables (exponential growth) has appeared in the literature of optimization. It might be worth mentioning this point. Also in the abstract, the author(s) stated that the exact convex dual problem can be solved in polytime. Is this statement really true given that you introduced exponentially many variables?
- In Theorem 2.2 (exact convex dual), our variables are $s_i$'s. How can we get the $W_i$'s from $s_i$'s (primal recovery)? This question is just from my curiosity, and it will not affect my rating given that this paper is studying the property of BN instead of proposing a practical algorithm.
- For the implicit regularization part. Assume that SGD with BN is converging to $w^*_{BN}$ and SGD without BN is converging to $w^*_{noBN}$. Then can the analysis in Section 4 give us the conclusion that $w^*_{BN}$ is ''better'' than $w^*_{noBN}$ according to some kind of regularization condition? The conclusion from Section 4 is a little bit vague and I think the presentation of that part can be improved.


[1] Santurkar et al. How Does Batch Normalization Help Optimization? NeurIPS 2018.

[2] Arora et al. Theoretical analysis of auto rate-tuning by batch normalization. ICLR 2019.

**Summary Of The Paper:**

The author(s) presented a quite comprehensive theoretical study of batch normalization from the lens of convex duality. Numerical experiments are also conducted to support the theoretical analysis.

**Summary Of The Review:**

Overall I think the results presented in this work are solid and interesting. There is some missing related work, but I think it is fine since the results in this paper have no significant overlap with the existing results. I did not check the proofs in this paper in detail.

---

### Official Review · Reviewer_syyN · 2021-11-06

**Correctness:** 2
**Technical Novelty And Significance:** 4
**Empirical Novelty And Significance:** 4
**Recommendation:** 5
**Confidence:** 3

**Main Review:**

It is crystal clear that a polynomial-time training algorithm for neural networks (with or without BN) is a breakthrough in machine learning. Yet, we need careful proof checking for this strong claim. I am lost in the notations and the proofs. I kindly request the author to help me to proof check the result.
1. What is the difference between $W^{(1)}$ and $w^{(1)}$ in equation 7? What are the dimensions of $y$? Why problem (7) is easier than (6) to solve? (7) has a non-convex constraint.
2. Intuitively, I do not understand why dual with respect to the output weights may make the original non-convex program easier. The original problem is convex w.r.t the output weights. We can compute the optimal output weights in closed form. But this is not helpful for convexifying the objective.
3. What authors do mean by $\mp$ in Theorem 2.1?
4. For a sanity check of Thm. 2.1, consider $\beta=0$. Using Lemma C.1., one can check that $d_2^*=0$. But, it is easy to design input for which $p_2^* >0$. Why does this sanity check fail? Did I misunderstand the results?
5. Does Theorem 2.1 imply that all the $w_j^{(1)}$ are equal?
6. Recall inequality $P = O(r (n/r)^r)$. If $r=n$, then $P=(n)$? I think this can not be true and $P$ is at least $O(2^n)$.
7. The proof of Thm 2.2. is not clear. Would authors please add the complete proof instead of "directly
follow the steps in Pilanci & Ergen (2020)".



**Summary Of The Paper:**

The claim of this paper is casting training neural networks with batch normalization to a convex program solvable in polynomial time. The convex reduction sparks an implicit regularization of batch normalization. Taking inspiration from the convex program and the implicit regularization, the authors improve BN.

**Summary Of The Review:**

If this result is correct, it would be a breakthrough in machine learning. Yet, we need to make sure the proofs are correct. The current version is a bit difficult to read and check. Notations are not introduced well and the proofs are referring to other recent results. I recommend authors make a self-contained paper.  I also need the author's responses on my failed sanity check to understand the proof better.

---

### Decision · Program_Chairs · 2022-01-20

**Decision:**

Accept (Poster)

**Comment:**

The authors rely on the recent convex formulation of neural network training to establish an interesting correspondence between whitening of input data and batch normalization (after and before relu). While convex formulation is not scalable to realistic architectures, it is a great insight that would add value to our understanding of the batch normalization tool.